# Strong Sensitivity of the Isotopic Composition of Methane to the Plausible Range of Tropospheric Chlorine

Sarah A. Strode[1,2,*], James S. Wang[1,2,**], Michael Manyin[2,3], Bryan Duncan[2], Ryan Hossaini[4], Christoph A. Keller[1,2], Sylvia E. Michel[5], James W. C. White[5]

[1]Universities Space Research Association, Columbia, MD, USA

[2]NASA Goddard Space Flight Center, Greenbelt, MD, USA

[3]SSAI, Lanham, MD, USA

[4]Lancaster Environment Centre, Lancaster University, Lancaster, UK

[5]Institute of Arctic and Alpine Research, University of Colorado, Boulder, CO, USA

[*]*correspondence to*: sarah.a.strode@nasa.gov

[**]Now at the Institute for Advanced Sustainability Studies, Potsdam, Germany

**Abstract.** The $^{13}$C isotopic ratio of methane, $\delta^{13}$C of $CH_4$, provides additional constraints on the $CH_4$ budget to complement the constraints from $CH_4$ observations. The interpretation of $\delta^{13}$C observations is complicated, however, by uncertainties in the methane sink. The reaction of $CH_4$ with Cl is highly fractionating, increasing the relative abundance of $^{13}CH_4$, but there is currently no consensus on the strength of the tropospheric Cl sink. Global model simulations of halogen chemistry differ strongly from one another in terms of both the magnitude of tropospheric Cl and its geographic distribution. This study explores the impact of the inter-model diversity in Cl fields on the simulated $\delta^{13}$C of $CH_4$. We use a set of GEOS global model simulations with different predicted Cl fields to test the sensitivity of the $\delta^{13}$C of $CH_4$ to the diversity of Cl output from chemical transport models. We find that $\delta^{13}$C is highly sensitive to both the amount and geographic distribution of Cl. Simulations with Cl providing 0.28% or 0.66% of the total $CH_4$ loss bracket the $\delta^{13}$C observations for a fixed set of emissions. Thus, even when Cl provides only a small fraction of the total $CH_4$ loss and has a small impact on total $CH_4$, it provides a strong lever on $\delta^{13}$C. Consequently, it is possible to achieve a good representation of total $CH_4$ using widely different Cl concentrations, but the partitioning of $CH_4$ loss between the OH and Cl reactions leads to strong differences in isotopic composition depending on which model's Cl field is used. Comparing multiple simulations, we find that altering the tropospheric Cl field leads to approximately a 0.5‰ increase in $\delta^{13}CH_4$ for each percent increase in how much $CH_4$ is oxidized by Cl. The geographic distribution and seasonal cycle of Cl also impacts the hemispheric gradient and seasonal cycle of $\delta^{13}$C. The large effect of Cl on $\delta^{13}$C compared to total $CH_4$ broadens the range of $CH_4$ source mixtures that can be reconciled with $\delta^{13}$C observations. Stronger constraints on tropospheric Cl are necessary to improve estimates of $CH_4$ sources from $\delta^{13}$C observations.

## 1. Introduction

The global budget of methane is of great interest due to methane's role as a greenhouse gas, ozone precursor, and sink of the hydroxyl radical. Despite extensive study, major uncertainties in the methane budget remain, with top-down and bottom-up estimates often yielding different results (Kirschke et al., 2013;Saunois et al., 2016;Saunois et al., 2017, and refs therein) for the strength of specific source types. Furthermore, the resumed increase of methane concentrations beginning in 2007 (Dlugokencky et al., 2009; Rigby et al., 2008) can be explained by multiple hypotheses including an increase in fossil fuel emissions (Turner et al., 2016;Thompson et al., 2015;Hausmann et al., 2016), an increase in fossil fuel emissions combined with a decrease in biomass burning (Worden et al., 2017), an increase in biogenic sources (Schaefer et al., 2016;Nisbet et al., 2016), or a decrease in hydroxyl concentrations (Turner et al., 2017;Rigby et al., 2017). Variations in hydroxyl concentrations may also be important for the decrease in methane growth from 1999-2006 (McNorton et al., 2016).

Observations and modeling of methane's carbon isotopes provides additional information on methane sources since individual sources differ in their $^{13}C$ to $^{12}C$ ratio ($\delta^{13}C$). Isotopic information can be used to better constrain methane sources (e.g. Thompson et al., 2015; Mikaloff Fletcher et al., 2004b, a) and infer how the source mixture changed over glacial (e.g. Hopcroft et al., 2018; Fischer et al., 2008; Bock et al., 2017), millennial (e.g. Ferretti et al., 2005; Houweling et al., 2008), and decadal timescales (e.g. Nisbet et al., 2016; Schaefer et al., 2016; Kai et al., 2011; Schwietzke et al., 2016; Thompson et al., 2018). However, there are considerable uncertainties in the processes that control methane's isotopic composition that may confound source apportionment studies. Many modeling studies use a single value for the isotopic ratio of each source, while in reality sources such as wetlands, biomass burning, and natural gas show large regional or environment-dependent variations in their isotopic signature (Ganesan et al., 2018; Brownlow et al., 2017; Dlugokencky et al., 2011; Schwietzke et al., 2016; Sherwood et al., 2017).

The isotopic composition of atmospheric methane is also sensitive to methane's sinks. Reaction with OH, the principal loss for atmospheric methane, has a kinetic isotope effect (KIE) of $-5.4‰$ ($\alpha=k_{13}/k_{12}=0.9946$) to $-3.9‰$ ($\alpha=0.9961$) (Saueressig et al., 2001; Cantrell et al., 1990) and contributes to the interhemispheric gradient of $\delta^{13}C$ (Quay et al., 1991). Mass balance (Lassey et al., 2007) and observations of the seasonal cycle of $\delta^{13}C$ versus methane concentration, however, suggest larger apparent KIE values, which may indicate a role for methane oxidation by chlorine (Cl) in the marine boundary layer (MBL) (Allan et al., 2001;Allan et al., 2007) since Cl has a KIE of $-61.9‰$ ($\alpha=0.938$) at 297K (Saueressig et al., 1995). Inclusion of the MBL Cl sink alters the source mixture inferred from inverse modeling of $\delta^{13}CH_4$ (Rice et al., 2016). Nisbet et al. (2019) point out that interannual variability in the $CH_4$ Cl sink could explain some of the variability of $\delta^{13}C$. Cl is also an important methane sink in the stratosphere, and the impact of this sink on surface $\delta^{13}C$ is a source of uncertainty in modeling $\delta^{13}C$ (Ghosh et al., 2015). Reaction with stratospheric Cl contributes approximately 0.23‰ to the $\delta^{13}C$ of surface methane and makes a small contribution to the observed trend in surface $\delta^{13}C$ over the last century (Wang et al., 2002).

The global concentration of Cl in the MBL and its role in the methane budget is still uncertain. Cl concentrations are highly variable and not well constrained by direct observations. Modeling work by Hossaini et al. (2016) and Sherwen et al. (2016) suggests that chlorine provides 2-2.5% of tropospheric methane oxidation. This

agrees well with estimates based on the isotopic fractionation, which also suggest Cl provides several percent of the
total sink (Allan et al., 2007;Platt et al., 2004). However, Gromov et al. (2018) suggest that these are overestimates
as values over 1% are inconsistent with the $\delta^{13}$C of CO, which is a product of $CH_4$ oxidation. The recent modeling
study of Wang et al. (2019) also suggests a value of 1%. There is thus considerable uncertainty in the role of chlorine
in the budget and isotopic composition of methane.
Here, we investigate the sensitivity of $\delta^{13}$C of $CH_4$ to inter-model diversity in tropospheric chlorine
concentrations to better quantify how much uncertainty in the interpretation of $\delta^{13}$C is imposed by the uncertainty in
Cl. Section 2 describes the modeling framework. We present results for total $CH_4$ and its isotopic composition
compared to surface observations in Section 3, and discuss the implications for the global $CH_4$ budget in Section 4.
**2. Methods**

**2.1 Model Description**

We simulate atmospheric methane with the Goddard Earth Observing System (GEOS) global earth system model
(Molod et al., 2015; Nielsen et al., 2017). The model has 72 vertical levels extending from the surface to 1 Pa. We
conduct simulations at C90 resolution on the cubed sphere, which corresponds to approximately 100 km horizontal
resolution. The simulations' meteorology is constrained to the MERRA-2 reanalysis (Gelaro et al., 2017) using a
"replay" method (Orbe et al., 2017). The GEOS replay agrees well with the tropospheric mean age of the Global
Modeling Initiative (GMI) chemistry and transport model (CTM) (Orbe et al., 2017), which shows reasonable
agreement with the age derived from $SF_6$ observations, albeit with an old bias in the southern hemisphere (Waugh et
al., 2013). We thus expect the simulated interhemispheric transport time to be reasonable.
The GEOS $CH_4$ simulation can be interactively coupled to CO and OH (Elshorbany et al., 2016), or run
independently with prescribed OH fields. We take the latter approach in this study, since this approach is able to
capture many of the observed variations in atmospheric methane (Elshorbany et al., 2016). We prescribe the OH field
following (Spivakovsky et al., 2000), but modify the OH to be approximately 20% higher in the Northern Hemisphere
than the Southern Hemisphere, consistent with the OH field produced by many global atmospheric chemistry models
(Naik et al., 2013;Strode et al., 2015). This modification is designed to make our results more applicable to
understanding the impacts of inter-model differences in Cl, since it makes our OH distribution more consistent with
that produced by many CCMs. The OH field varies monthly but repeats every year. We also include stratospheric
losses for $CH_4$ from reaction with OH, Cl, and $O^1D$. These fields are prescribed from output of the GMI CTM
(https://gmi.gsfc.nasa.gov) (Strahan et al., 2007; Duncan et al., 2007).
We implement the $CH_4$ isotopes in GEOS by separately simulating $^{13}CH_4$ and $^{12}CH_4$ tracers. We then calculate
total $CH_4$ as the sum of the two carbon isotopologues and calculate $\delta^{13}$C of $CH_4$ in per mil using the standard definition:
$\delta^{13}$C-$CH_4$ (‰) = ([$^{13}CH_4$]/[$^{12}CH_4$]/$R_{std}$ – 1) * 1000         (1)
where $R_{std}$=0.0112372 is the peedee belemnite isotopic standard (Craig, 1957). We partition each emission source
into $^{12}CH_4$ and $^{13}CH_4$ emissions according to a source-specific $\delta^{13}$C value from the literature, provided in Table 1. We
use the Craig (1957) $R_{std}$ value to partition the sources since it is cited in the literature used in Table 1 (Houweling et
al, 2000; Lassey, 2007), and so for consistency we use the same value in equation 1 to calculate the simulated $\delta^{13}C$ of
the $CH_4$ concentrations. We note, however, that the GMD observations now use a slightly different standard, the
VPDB value of 0.011183 (Zhang and Li, 1990). A sensitivity study (not shown) confirms that the choice Rstd has
little effect on our results as long as the same value is used for the source partitioning as for the calculation of $\delta^{13}C$-
$CH_4$ from simulated $[^{13}CH_4]$ and $[^{12}CH_4]$.

The reaction rates for $CH_4+OH$, $CH_4+Cl$, and $CH_4+O^1D$ differ between the $^{12}CH_4$ and $^{13}CH_4$ simulations to

account for the kinetic isotope effect (KIE). In particular, we assume $\alpha$ values of 0.987 and 0.938 for $CH_4+O^1D$ and
$CH_4+Cl$, respectively (Saueressig et al., 1995;Saueressig et al., 2001). Our standard simulation uses $\alpha_{OH} = 0.9946$
(Cantrell et al., 1990).

Methane from different sources is tracked individually using a "tagged tracer" approach, which allows us to

simulate the spatial footprint of $CH_4$ and $\delta^{13}C$-$CH_4$ from individual sources. The soil sink is applied to each tracer as
a fraction of its source, modified to account for faster loss of $^{12}CH_4$ to soil compared to $^{13}CH_4$ ($\alpha_{soil} = 0.978$) (Tyler et
al., 1994). Supplemental figure S1 shows the July 2004 $CH_4$ and $\delta^{13}C$-$CH_4$ footprints of the biomass burning, wetland,
and coal + other geologic $CH_4$ sources from the tagged tracers to illustrate the tagged tracer approach. We note that
the $\delta^{13}C$ values of the surface methane from each source is heavier (less negative) than the emission value for that
source (Table 1), especially in regions far from the source, because of the fractionating effects of the sinks.
Supplemental Fig. S2 shows the corresponding footprints for January.
**2.2      Description of Simulations**

We simulate the period from 1990 through 2004, and focus our analysis on 2004. We choose 2004 as our endpoint

because it lies within the period when methane concentrations remained relatively flat, simplifying our analysis.
Ending the simulations in 2004 also avoids much of the uncertainty about the causes of the resumed growth rate in
recent years. The isotopic ratios of methane take longer to adjust to a perturbation than total methane (Tans, 1997).
Since we wish to begin our simulations with a state that is as close as possible to "spun up", we choose the initial
condition for each tagged tracer based on its present-day distribution and proportion of the total $CH_4$ and scale it back
to 1990 levels such that the total $CH_4$ is consistent with the global mean $CH_4$ from surface observations for 1990. We
then iteratively adjusted the $^{12}C$- to $^{13}C$-$CH_4$ tracer ratios at the beginning of 1990 to yield a good match to global
mean $\delta^{13}C$-$CH_4$ observations for 1998, when more $\delta^{13}C$-$CH_4$ observations are available. The same initial condition is
used for the standard and sensitivity simulations.

We use interannually-varying emissions of $CH_4$ from anthropogenic, biomass burning, and wetland sources.

Emissions from anthropogenic sources such as oil and gas, energy production, industrial activities, and livestock come
from the EDGAR version 4.2 inventory (European Commission, 2011). Biomass burning emissions come from the
MACCity inventory (Granier et al., 2011). We treat forest fires as C3 burning and savannas as C4 burning for
partitioning the biomass burning emissions between isotopologues. Wetland and rice emissions come from the
Vegetation Integrative Simulator for Trace gases (VISIT) terrestrial ecosystem model (Ito and Inatomi, 2012), scaled
by 0.69 and 0.895, respectively, for consistency with the Transcom-$CH_4$ study (Patra et al., 2011). Ocean (Houweling
et al., 1999), termite (Fung et al., 1991), and mud volcano emissions (Etiope and Milkov, 2004) are also from the
Transcom study (Patra et al., 2011) and have a seasonal cycle but no interannual variability. Initial tests with these
emissions showed a substantial underestimate of the $CH_4$ growth rate. Consequently, we scale up all the emissions
by 10% for 1990-1998, and by 6.8% for 1998-2004. We find the resulting emissions lead to a good simulation of the
timeseries of surface $CH_4$ observations from the National Oceanic and Atmospheric Administration (NOAA) Global
Monitoring Division (GMD) (Dlugokencky et al., 2018), especially towards the end of the period (Fig. 1). The
simulation has only a 0.1% mean bias compared to the observations for 2004.
Our standard simulation (SimStd) uses Cl from the GMI CTM for the tropospheric as well as stratospheric loss
of $CH_4$ by reaction with Cl. Tropospheric Cl concentrations are small in GMI since it does not include very short-
lived species, and reaction with Cl represents only 0.28% of the total tropospheric $CH_4$ loss. We also conduct several
sensitivity simulations in which we alter the tropospheric and lower stratospheric Cl fields (Table 2). Cl is not altered
above 56 hPa. Sensitivity simulation SimGC uses Cl from the GEOS-Chem chemistry module within GEOS (Long
et al., 2015; Hu et al., 2018). GEOS-Chem v11-02f with fully coupled tropospheric and stratospheric chemistry was
used for this simulation, with halogen chemistry as described in Sherwen et al. (2016). SimGC has higher values of
tropospheric Cl than SimStd (Figs. 3,4) and leads to 0.66% of the total $CH_4$ loss occurring via Cl. Both SimStd and
SimGC are thus below the 1% loss via Cl suggested by (Gromov et al., 2018). We conduct a third sensitivity
simulation, SimTom, which uses Cl from the TOMCAT model simulations that include chlorine sources from
chlorocarbons (including very short-lived substances), HCl from industry and biomass burning, and very short lived
substances (Hossaini et al., 2016). This simulation leads to Cl accounting for 2.5% of tropospheric $CH_4$ loss in our
simulation. Finally, we conduct a fourth sensitivity simulation, SimMBL, which modifies the Cl over the oceans at
altitudes below 900 hPa (Fig. 2d) to reflect the marine boundary layer distribution suggested by (Allan et al., 2007).
This Cl field is described by the following equation:
$Cl\_MBL = 18*10^3$ atoms/$cm^3$ * $(1 + \tanh(3\lambda)*\sin(2\pi*(t-90)/365))$  (2)
where $\lambda$ is latitude in radians and t is the day of the year. Elsewhere SimMBL uses the Cl field from SimStd. This
simulation has the highest percent of $CH_4$ loss occurring via Cl: 3.9%. If we consider the loss of methane throughout
the atmosphere rather than just the troposphere, then the percent lost via Cl increases to 1.6%, 2.0%, 3.6% and 5.0%
for SimStd, SimGC, SimTom, and SimMBL, respectively.

We designed the sensitivity experiments to alter the isotopic composition of $CH_4$ without greatly affecting

the total $CH_4$. Consequently, we reduce the OH concentrations in the SimTom and SimMBL simulations by 2% and
4%, respectively, relative to the SimStd OH to offset the effect of increasing Cl. These changes are small compared
to the uncertainty in global OH (Rigby et al., 2017). In addition, the SimTom and SimMBL simulations use
$\alpha_{OH}$=0.9961 (Saueressig et al., 2001) rather than $\alpha_{OH}$=0.9946 (Cantrell et al., 1990) to avoid too much fractionation
from the combined Cl and OH sinks. While these changes are necessary to maintain consistent total $CH_4$ and
reasonable isotopic ratios, changing multiple factors in addition to Cl makes it difficult to quantify the impact of Cl
alone. Consequently, we conduct an additional sensitivity study, called SimTomB, which uses the same Cl field as
SimTom but retains the OH and $\alpha_{OH}$ values of SimStd. SimTomB is used in Section 3.3. This simulation becomes
too heavy compared to observations, justifying the need to change $\alpha_{OH}$ in the main SimTom simulation. We also
conduct a sensitivity simulation, SimOHp, that uses the same Cl field as SimStd but does not alter the hemispheric
ratio of OH. Table 2 summarizes the standard and sensitivity simulations.
The four Cl distributions differ in their vertical and horizontal spatial distributions as well as their
tropospheric mean (Figs. 2 and 3). The SimStd Cl is largest in the tropics, nearly symmetric between hemispheres,
and increases with altitude. Both SimGC and SimTom have Cl that is larger in the Northern Hemisphere than the
Southern Hemisphere in the annual mean and reaches a minimum in the mid-troposphere. However, the maximum in
lower tropospheric Cl occurs in the tropics in SimGC but in the extratropics in SimTom. This mid-latitude Cl
maximum arises because SimTom has high Cl values over east Asia, whereas SimGC Cl is highest over ocean regions
(Fig. 3). SimMBL has a strong maximum in the MBL compared to the free troposphere and land regions. Its annual
mean Cl concentrations are higher in the Southern Hemisphere (Fig. 2) due to the larger ocean area in the Southern
Hemisphere. However, SimMBL includes a strong seasonal shift in peak Cl between the hemispheres. SimStd and
SimGC have more modest seasonal shifts, while Cl in SimTom remains concentrated in the northern hemisphere
throughout the year (Fig. S3). All simulations repeat the same Cl field from year to year.
The sensitivity simulations listed above are designed to test the role of the Cl sink. We conduct an additional
sensitivity study, SimWet, to illustrate the role of spatial variation in the isotopic source signature. SimWet parallels
SimStd, but the isotopic composition of the wetland source uses spatial variation from Ganesan et al (2018). The
global mean source signature of the wetland emissions remains -60‰.
**2.3    Observations**
We use surface observations from the NOAA GMD Carbon Cycle Cooperative Global Air Sampling Network to
evaluate our simulations. We use the monthly mean observations of total $CH_4$ (Dlugokencky et al, 2018) and $\delta^{13}C$ of
$CH_4$ (White et al., 2018) to compare to the monthly mean simulation results. The isotopic measurements were made
at the Institute of Arctic and Alpine Research at the University of Colorado and are referenced to the VPDB scale
(Zhang and Li, 1990). The analytical uncertainty of the isotopic measurements is 0.06‰. The variability between
measurements taken in a given month may, however, be larger, so we use the maximum of analytical uncertainty and
the within-month standard deviation as the uncertainty in the monthly mean. When multiple years are observations
are averaged together, we use the pooled variance to calculate the standard error, thus reducing the error based on the
number of years. The GMD observations are located at remote sites, shown in Fig. 4 for $CH_4$ in 2004. Measurements
of $\delta^{13}C$ of $CH_4$ are available at a subset of the sites, shown in Fig. 5.
**3.   Results and Discussion**
**3.1 Evaluation of Simulated $CH_4$**
We find good agreement between the SimStd simulation and the GMD observations for $CH_4$ (Fig. 4) for 2004.
We focus on these two months to represent the seasonal differences. The latitudinal distribution is well-reproduced,
and the simulation captures the elevated concentrations of $CH_4$ observed over Europe in January as well as the January
versus July differences in concentration. Overall, the spatial correlation between SimStd and the observations is 0.93
in January and 0.85 in July. The sensitivity simulations described in Table 2 have little effect on the $CH_4$ distribution,
as shown by the overlapping symbols in Fig. 4c,d.

**3.2 Impact of Cl on the $\delta^{13}C$ Distribution**

We next examine the distribution of $\delta^{13}C$ in SimStd compared to observations. Figure 6 shows the timeseries of

observed and simulated $\delta^{13}C$ for 1998-2004 at the 6 GMD sites with $\delta^{13}C$ records covering this time period. We begin
the figure at 1998 rather than 1990 due to the lack of data availability in the earlier years. The standard and sensitivity
simulations overestimate $\delta^{13}C$ at the northernmost station, BRW. The observations at the other stations lie within the
range of simulations, with most simulations underestimating the observations at the south pole. The differences
between the different sensitivity simulations are large compared to the interannual variability in both observed and
simulated $\delta^{13}C$. We focus our subsequent analysis on a single year, 2004.

Fig. 5a,b shows both meridional and zonal variability in $\delta^{13}C$. Background values are less negative (heavier) in

the Southern versus Northern Hemisphere (NH) (Fig. 7), a feature seen more strongly in the observations, but there is
also variability due to the different source signatures. Areas of biomass burning, such as Tropical Africa, show up as
particularly heavy, while regions with large wetland and rice emissions, such as SE Asia, are particularly light.
Another prominent feature is the isotopically heavy region in northern Eurasia (around 60°N) in January, which we
attribute to the influence of the geologic (including oil, gas, and coal) source in this region (Supp. Fig. S2). This signal
is less evident in July, when greater influence from boreal wetlands lightens the isotopic mix. The spatial correlation
($r^2$) between the SimStd and observed $\delta^{13}C$ is 0.61 in January and 0.75 in July.

The sensitivity simulations with altered oxidant concentrations alter the global values of $\delta^{13}C$, but the geographic

patterns remain similar to that of SimStd. The larger Cl sink in SimGC leads to an overall less negative $\delta^{13}C$, which
agrees better than SimStd with observations at Southern Hemisphere (SH) sites but worse in the NH (Figs. 6c,d and
7). The isotopic effect of the larger Cl sink in SimTom is compensated by the lower OH and $\alpha_{OH}$ values used in that
simulation, flattening the interhemispheric gradient (Figs. 6e,f and 7). In contrast, the very large MBL Cl
concentrations in SimMBL lead to an overestimate (insufficiently negative) of the observed $\delta^{13}C$ (Fig. g,h), but
strengthens the interhemispheric gradient. We note that since all simulations began with the same initial conditions
but have different sinks, the isotopic composition is not in steady state in 2004 and the results of the sensitivity
simulations diverge further with additional years of simulation, with SimMBL becoming clearly inconsistent with
observations. We note that while these results highlight the differences in $\delta^{13}C$ imposed by changing Cl, the absolute
values of $\delta^{13}C$, and hence their agreement with observations, would be different for $CH_4$ source mixtures with a
different average $\delta^{13}C$.

Figure 7 reveals an underestimate in the interhemispheric gradient of $\delta^{13}C$ in both SimStd and the sensitivity runs

compared to the GMD observations. Table 3 presents the observed and simulated $\delta^{13}C$ interhemispheric gradients
calculated as the difference between the $\delta^{13}C$ values averaged over all sites south of 30°S and the average over sites
north of 30°N. SimStd and SimGC show similar underestimates of the observed gradient, and the underestimate is
more severe in SimTom. The gradient is improved in SimMBL in January. The differences between simulations
reflect differences in the locations where CH4 oxidation occurs and the amount and location of isotopic fractionation
due to Cl versus OH.  Fig. 8 shows that the higher Cl values over the NH, particularly China, in SimTom versus
SimStd leads to more CH4 loss occurring in the NH and higher (heavier) $\delta^{13}C$ in the NH.  This effect is particularly
pronounced over China and Europe.  Less fractionation by the OH sink in SimTom leads to lighter values in the SH.
Conversely, SimMBL has more loss occurring over the SH oceans in January, leading to heavier $\delta^{13}C$ in the SH (Fig.
9).  This effect is not present in July, when the SimMBL Cl loss shifts to the NH (Fig. S4).  The reduced hemispheric
difference in OH in SimOHp leads to a small improvement in the hemispheric gradient in $\delta^{13}C$.
We further examine the seasonal cycle of $\delta^{13}C$ in Fig. 10.  We focus on the seasonal cycle at the South Pole
Observatory (SPO) site because it is far from large CH4 sources and thus the seasonal cycle depends strongly on the
seasonality of the CH4 sinks.  While all simulations lie mostly within the error bars of the observations, SimMBL has
the largest seasonal cycle amplitude, overestimating the seasonal cycle at of the SPO observations with a $\delta^{13}C$ value
that is both too heavy in Feb.-June and too light in Aug.-Nov.  In contrast, SimStd and the other sensitivity simulations
underestimate the magnitude of the observed seasonal cycle at SPO.  Supplemental Fig. S5 shows a large enhancement
in the seasonal cycle amplitude between SimMBL and the other simulations for the Cape Grim site in Tasmania
(CGO), but only a small change at other sites.  This suggests that while MBL Cl is attractive as an explanation for the
SH seasonality of $\delta^{13}C$, this explanation may be inconsistent with the inclusion of non-marine Cl sources.  However,
since the seasonal cycle amplitude at SPO lies in between SimMBL and the other simulations, it is possible that at an
MBL Cl source similar to that of SimMBL but with a smaller average value could reproduce the amplitude well.

## 3.3 Quantifying the Sensitivity of $\delta^{13}C$ to CH4 Loss by Cl


Given the substantial range in estimates for how much methane is lost by reaction with tropospheric Cl, it is
important to quantify the sensitivity of global mean surface $\delta^{13}C$ to the CH4 loss by Cl.  This analysis summarizes
the global impact of the isotopic effect of the Cl differences between simulation discussed above.   Fig. 11 shows the
global mean, area weighted surface $\delta^{13}C$ in 2004 as a function of the percent of CH4 oxidized by Cl for SimStd,
SimGC, and SimTomB the three simulations with the same OH and emissions but different Cl.  A strong linear
relationship is evident between the oxidation by Cl and the surface $\delta^{13}C$.  The slope of the linear regression line
indicates the expected increase in surface $\delta^{13}C$ for a change in the percent of CH4 oxidized by Cl.  Based on this
analysis we expect that surface $\delta^{13}C$ will increase by approximately 0.5‰ for each one % increase in CH4 loss by
Cl.

## 3.4 Sensitivity of $\delta^{13}C$ to the Isotopic Distribution of Sources


Other factors in addition to the Cl distribution likely contribute to the mismatch between the observed and
simulated interhemispheric gradients.  Fig. 5 shows the impact of the geologic source on the $\delta^{13}C$ values over northern
Asia.  A bias in either the strength or the isotopic composition of this source will impact the interhemispheric gradient.
Another likely contributing factor is our use of a globally uniform isotopic ratio for each source type.  Ganesan et al.
(2018) developed a global map of the isotopic signatures of wetland emissions. We use this map to impose spatially
varying isotopic ratios on our SimWet simulation. SimWet increases the amplitude of the seasonal cycle in $\delta^{13}$C-CH$_4$
particularly for northern latitudes sites such as ALT, BRW, and MHD (Supplemental Fig. S5). It has little effect on
the seasonal cycle at the SH CGO and SPO sites, where SimMBL shows a large effect on the cycle. SimWet results
in improved agreement with the observed interhemispheric gradient (Figs. 5,7; Table 3). SimWet is better able to
simultaneously match the $\delta^{13}$C-CH$_4$ observations at both the northernmost (BRW) and southernmost (SPO) sites
shown in Fig. 6 than the other simulations, even though all simulations reproduce the latitudinal distribution of CH$_4$
well (Fig. 4). This highlights the importance of spatially varying isotopic ratios for the $\delta^{13}$C-CH$_4$ distribution. The
size of the effect of including spatially varying ratios in wetland emissions depends on the strength of the wetland
emissions as well as the other sources. Including spatially-varying isotopic signature for other sources as well could
further modify the simulated interhemispheric gradient, potentially correcting some of the flat gradient of e.g. the
SimTom simulation.

**4.  Conclusions**

The role of Cl as a methane sink is a significant uncertainty in the global CH$_4$ budget, particularly with respect to
isotopes. The global distribution of Cl is not well known from observations, and the Cl distributions simulated by
global models varies widely from model to model. We investigated the sensitivity of the surface $\delta^{13}$C distribution of
CH$_4$ to the inter-model diversity in tropospheric Cl using a series of sensitivity studies with a global 3D model. Given
the uncertainties in CH$_4$ sources and their isotopic ratios, it is not possible to conclude from this study which Cl field
is best. However, the differences between the simulations provides insight on the strong lever that tropospheric Cl
exerts on the $\delta^{13}$C distribution.
Our standard and sensitivity simulations all reproduce well the geographic distribution of and temporal evolution
of CH$_4$ observed at the GMD surface sites. However, imposing Cl distributions from a range of chemical transport
models used in the scientific community leads to large differences in the simulated distribution of the $\delta^{13}$C of CH$_4$.
The CH$_4$ sinks from Cl in our SimStd and SimGC simulations are both below 1% of the total CH$_4$ sink, as suggested
by Gromov et al. (2018). Yet the SimStd and SimGC simulations underestimate and overestimate, respectively, the
observed $\delta^{13}$C in 2004, despite the fact that both include only a relatively small CH$_4$ sink from Cl.
Our ability to reproduce the observed latitudinal distribution of $\delta^{13}$C depends not only on the assumed value of
global mean Cl, but also its geographic distribution. The detailed halogen chemistry model (TOMCAT) of Hossaini
et al. (2016) places the maximum Cl values in the continental NH, in contrast to the large MBL Cl sink used in Allan
et al. (2007) to explain SH observations. We find that the strong NH Cl maximum, along with the resulting reduction
in OH fractionation required to maintain consistency with observations, acts to flatten the interhemispheric gradient
of $\delta^{13}$C, while the MBL Cl sink increases the hemispheric differences in NH winter and also strengthens the seasonal
cycle. However, the interhemispheric gradient is also influenced by spatial variation in the isotopic signatures of the
sources and uncertainties in the soil sink, complicating this issue.

Two values for the fractionating effect of OH ($\alpha_{OH}$) on $\delta^{13}C$ (Cantrell et al., 1990; Saueressig et al., 2001) are

widely cited in the literature. Combining the TOMCAT Cl fields with the $\alpha_{OH}$ of Saueressig et al. (2001) leads to an
underestimate of observed $\delta^{13}C$, but combining it with the Cantrell et al. (1990) $\alpha_{OH}$ would lead to an overestimate.
Reducing uncertainty in the fractionating effect of OH would thus improve our ability to constrain the role of Cl.

Observations of the $\delta^{13}C$ of $CH_4$ provide an important tool for constraining the $CH_4$ budget. We find that the

range of Cl fields available from current global models leads to a wide range of simulated $\delta^{13}C$ values. Each percent
increase in the amount of $CH_4$ loss occurring by reaction with Cl increases global mean surface $\delta^{13}C$ of $CH_4$ by
approximately 0.5‰. This relationship can be used to estimate the impact on methane's isotopic values from future
model simulations of Cl. The choice of Cl field thus strongly impacts what $CH_4$ source mixture best fits $\delta^{13}C$
observations. Better quantification of the role of Cl in the methane budget and further developing models of
tropospheric halogens is therefore critical for interpreting the $\delta^{13}C$ observations to their fullest potential.

**Data Availability**
The methane and $\delta^{13}CH_4$ observations are available from the NOAA GMD website:
https://www.esrl.noaa.gov/gmd/dv/data/. Output from the GEOS model is on the NASA Center for Climate
Simulation (NCCS) system.

**Author Contributions**
SS designed and conducted simulation, performed analysis, and prepared the manuscript. JSW contributed to model
development and experiment design. MM contributed to model development. BD contributed to model development
and conceptualization. RH and CK contributed inputs to the simulations. SM and JWCW contributed data and aided
in its interpretation. All authors contributed to the editing and revising of the manuscript.

The authors declare no competing interests.

**Acknowledgements**
Computational resources were provided by the NASA Center for Climate Simulation (NCCS). The authors thank
Prabir Patra for useful discussions. RH is supported by a NERC Independent Research Fellowship (NE/N014375/1).

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

**Table 1**: Emission source references and $\delta^{13}C$ values

| Source | Reference | IAV | $\delta^{13}C$ (‰)[a] | CH4 Source (Tg yr$^{-1}$)[b] |
|---|---|---|---|---|
| Animals (enteric fermentation) | EDGAR | Y | -62 | 102 |
| C3 Biomass Burning (Forests) | MACCity | Y | -26 | 16 |
| C4 Biomass Burning (Savannas) | MACCity | Y | -15 | 10 |
| Coal, energy, and industry | EDGAR | Y | -35 | 6 |
| Geologic (oil/gas/non-coal fuels, volcanos) | EDGAR, Transcom | Y, except volcanos | -40 | 124 |
| Waste (solid and animal waste, wastewater) | EDGAR | Y | -55 | 74 |
| Ocean | Transcom | N | -59 | 8 |
| Rice | Visit model | Y | -63 | 44 |
| Termites | Transcom | N | -57 | 22 |
| Wetlands | Visit model | Y | -60 | 149 |

[a]$\delta^{13}C$ values from Dlugokencky et al., 2011;Lassey et al., 2007;Monteil et al., 2011;Houweling et al., 2000 and refs
therein
[b]Values for 2004
**Table 2**: Oxidants for the Standard and sensitivity simulations

| Simulation | $[Cl]_{Trop}$[a] (molec cm$^{-3}$) | Cl Model[b] | Cl Reference | OH modification[c] |
|---|---|---|---|---|
| SimStd | 210 | GMI | (Strahan et al., 2007; Rotman et al., 2001; Strahan et al., 2013;Duncan et al., 2007) | $\alpha$ = 0.9946 |
| SimGC | 384 | GEOSChem | (Sherwen et al., 2016) | $\alpha$ = 0.9946 |
| SimTom | 1710 | TOMCAT | (Hossaini et al., 2016) | -2% [OH] $\alpha$ = 0.9961 |
| SimTomB | 1710 | TOMCAT | (Hossaini et al., 2016) | $\alpha$ = 0.9946 |
| SimOHp | 210 | GMI | See SimStd | Not modified for 20% higher in NH |
| SimMBL | 2810 | Tanh function below 900hPa over ocean; GMI elsewhere | (Allan et al., 2007) | -4% [OH] $\alpha$ = 0.9961 |

[a]Concentration of Cl averaged over the troposphere
[b]Name of the model that generated the offline Cl field
[c]Changes to [OH] or $\alpha_{OH}$ compared to SimStd
**Table 3:** Observed and Simulated Interhemispheric Gradient in $\delta^{13}C$-CH4

| | Jan. Gradient (‰)[a] | July Gradient (‰)[a] |
|---|---|---|
| GMD Obs | 0.36 | 0.28 |
| SimStd | 0.17 | 0.11 |
| SimGC | 0.17 | 0.098 |
| SimTom | 0.051 | 0.010 |
| SimMBL | 0.30 | 0.13 |
| SimOHp | 0.22 | 0.15 |
| SimWet | 0.28 | 0.25 |

[a]Average $\delta^{13}C$-CH4 at GMD site locations south of 30°S minus average $\delta^{13}C$-CH4 at locations north of 30°N

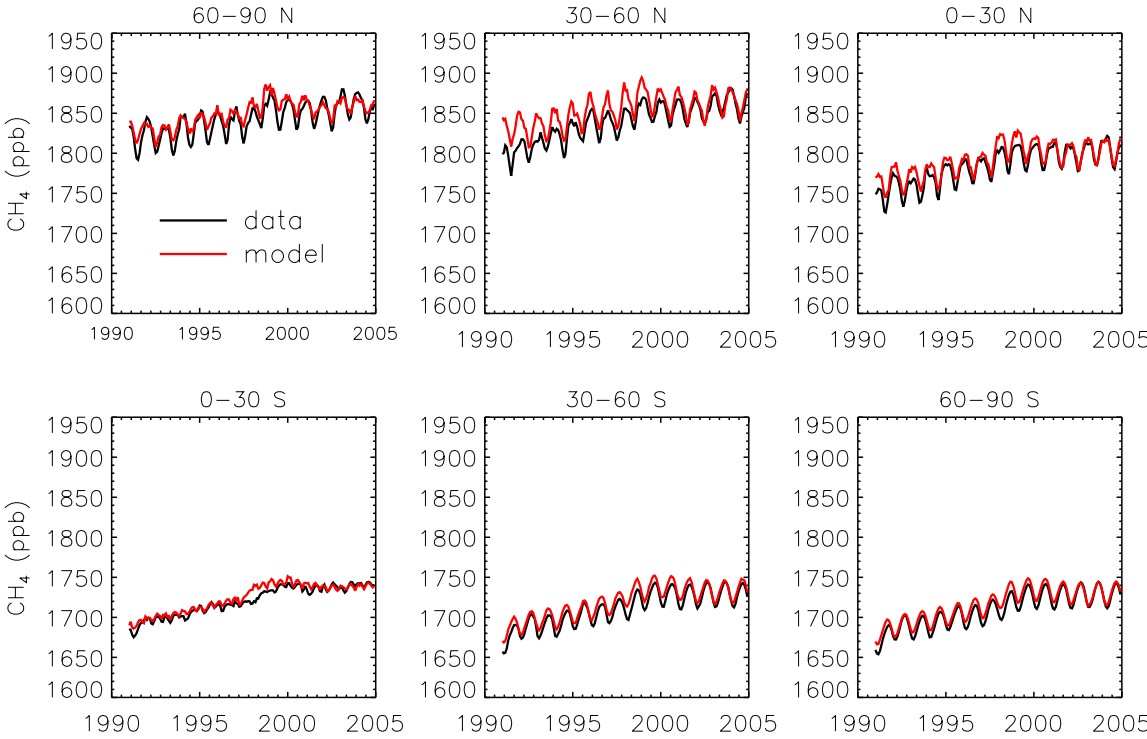

**Fig 1:** Monthly $CH_4$ observations from the GMD network (black) and simulated surface concentrations from SimStd
(red) averaged over latitude bands

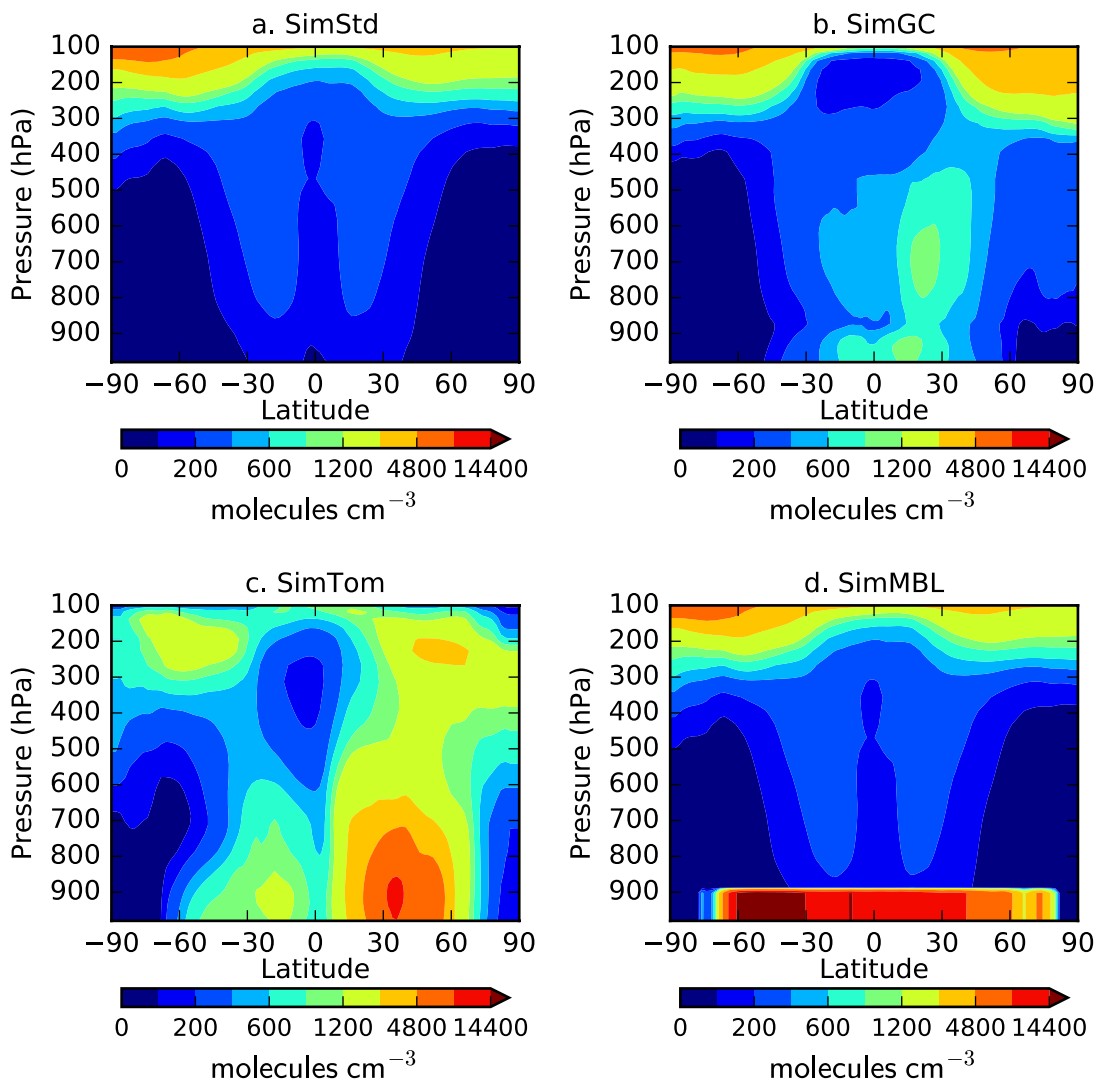

**Fig. 2**: Annual zonal mean Cl field for a) SimStd, b) SimGC, c) SimTom, and d) SimMBL.

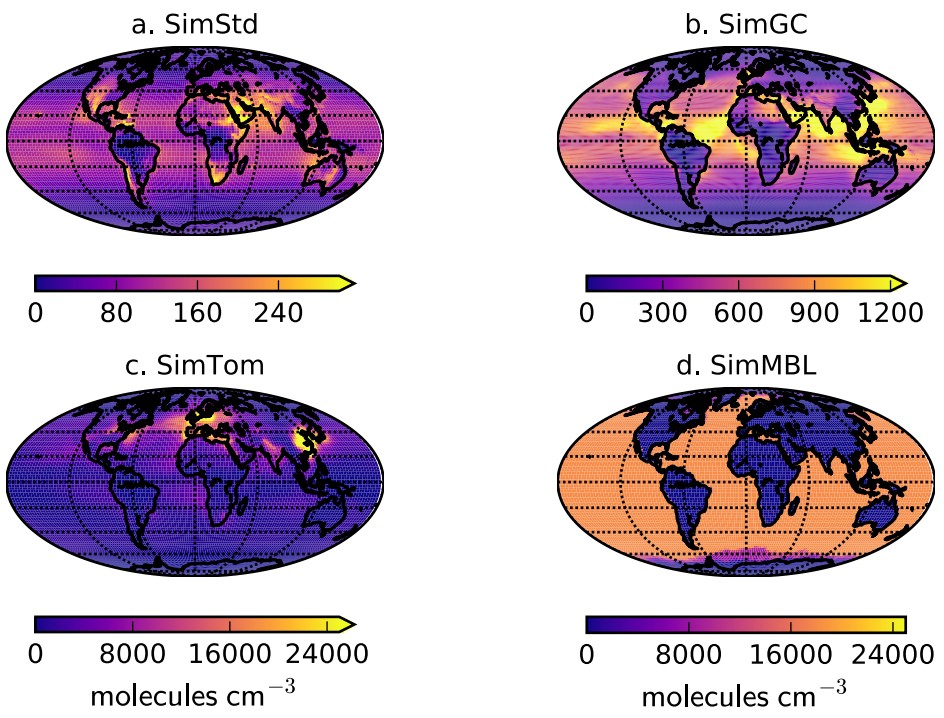

**Fig. 3:** Annual mean surface concentrations of Cl in a) SimStd, b) SimGC, c) SimTom, and d) SimMBL.  Note the
different color scales between panels.

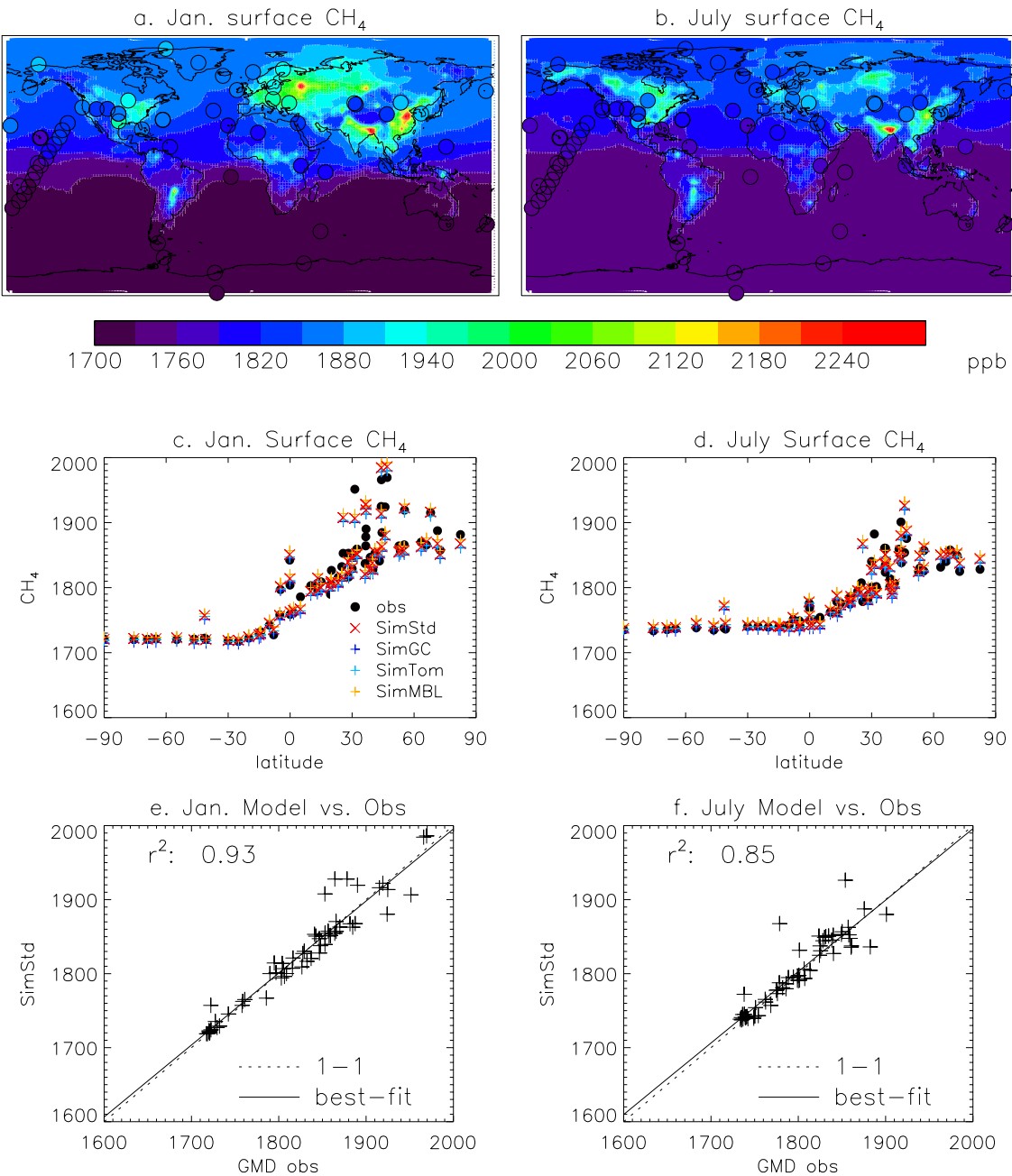

**Fig. 4**: Comparison of 2004 simulated and observed surface CH4 concentrations for January (left) and July (right). a,b) Surface concentrations of CH4 from SimStd are overplotted with the concentrations from the GMD observations in circles. c,d) GMD observations (black circles), SimStd (red x), SimGC (dark blue +), SimTom (light blue +), and SimMBL (orange +) CH4 as a function of latitude. E,f) SimStd CH4 (ppb) at the observation locations versus the GMD observations (+ signs) as well as the regression line (solid) and 1 to 1 line (dashed).

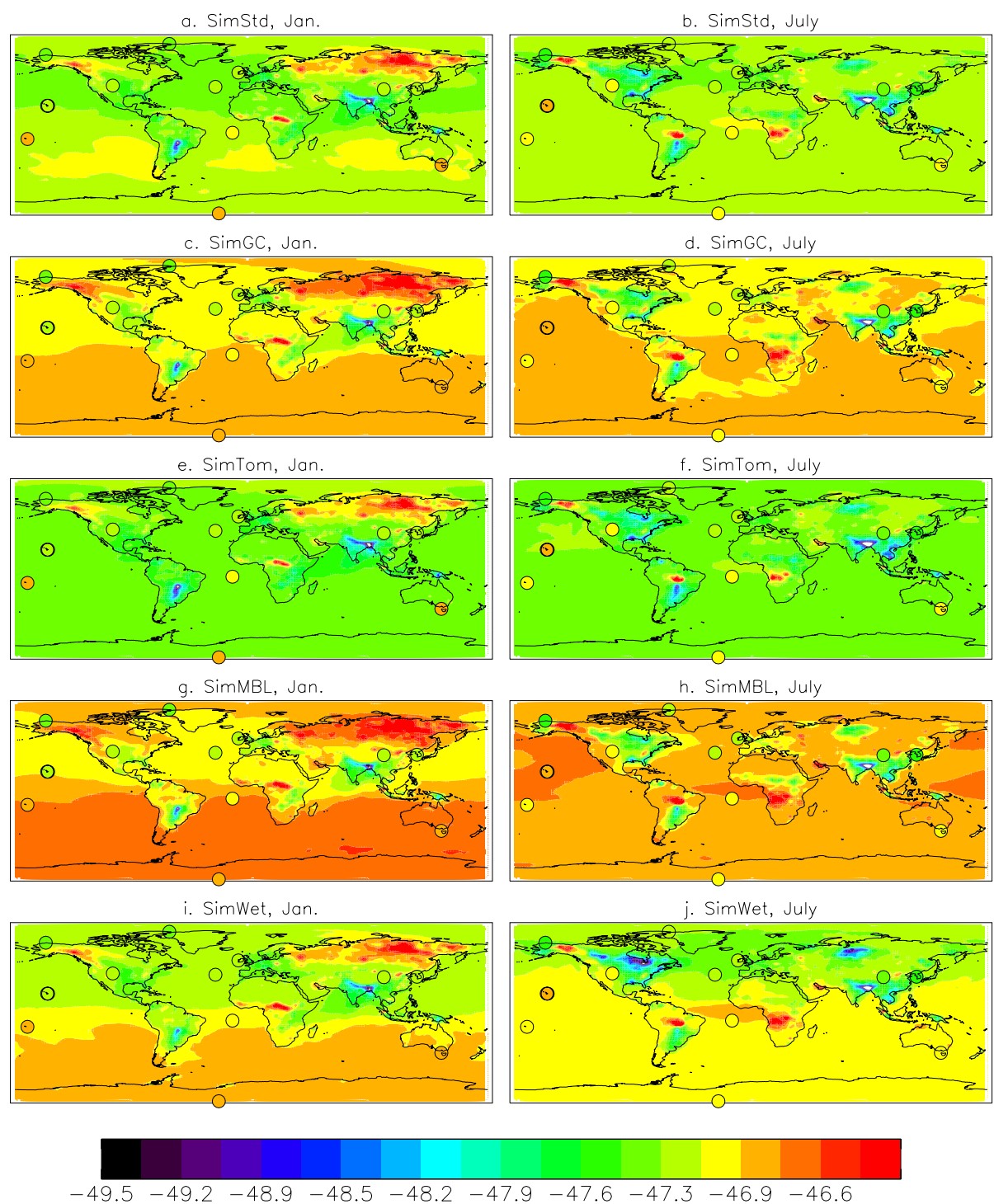

**Fig. 5:** Maps of the simulated surface $\delta^{13}C$ of $CH_4$ in per mil for Jan. (left) and July (right) overplotted with
observations from the GMD sites (circles). The simulations are (a,b) SimStd, (c,d) SimGC, (e,f) SimTom, (g,h)
SimMBL, and (I,j) SimWet.

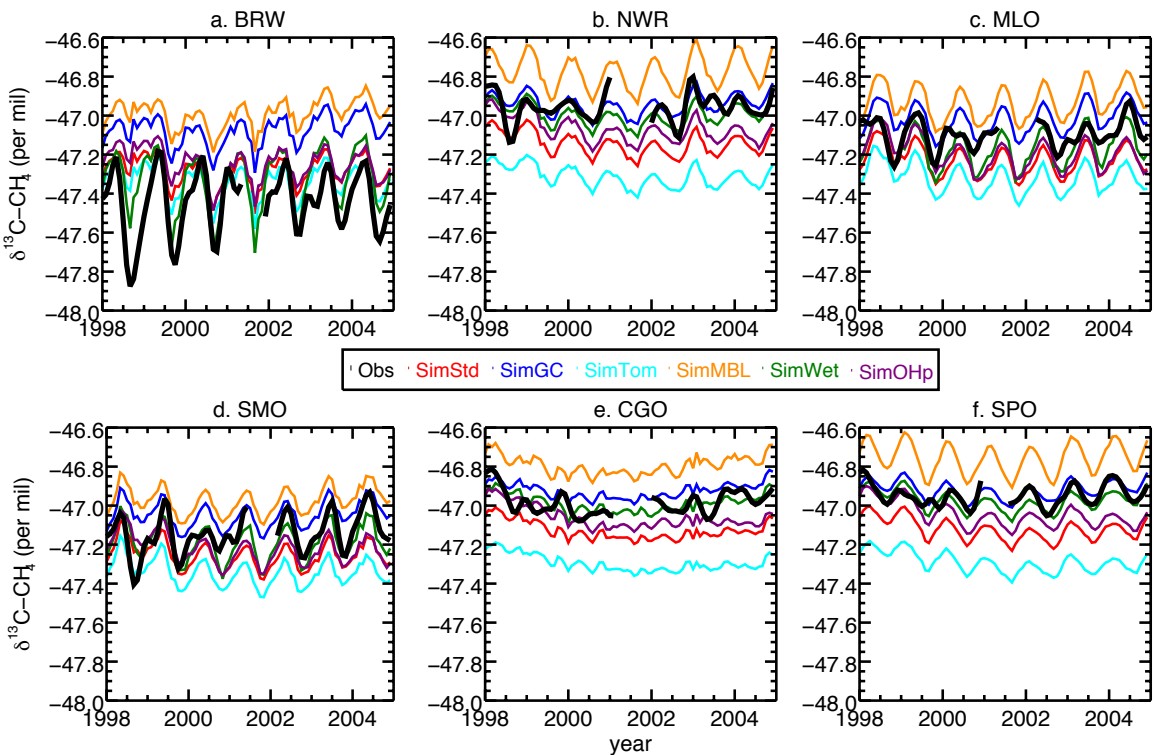

Fig. 6: The timeseries of observed (black) and simulated (colors) $\delta^{13}CH_4$ at the 6 GMD sites with records extending
back to 1998. BRW: 71.3°N, 156.6°W; NWR: 40.0°N, 105.6°W; MLO: 19.5°N, 155.6°W; CGO: 40.7°S, 144.7°E
and SPO: 90.0°S, 24.8°W.

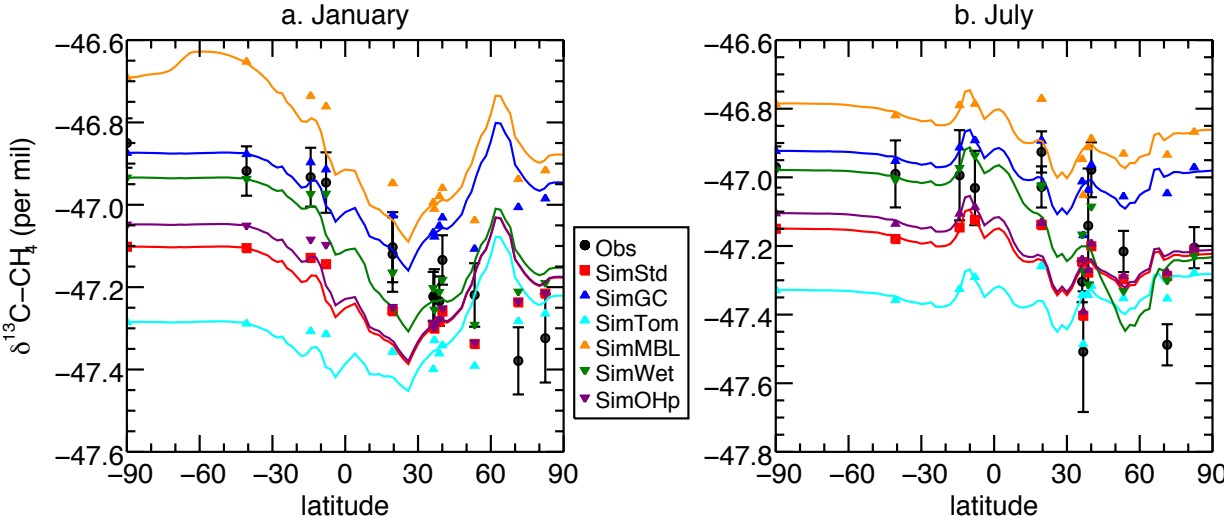

**Fig. 7:** $\delta^{13}C$ of CH$_4$ as a function of latitude in a) January and b) July 2004 for the GMD observations (Black
circles), SimStd (red), SimGC (dark blue), SimTom (cyan), SimMBL (orange), SimWet (green), and SimOHp
(purple). Errorbars represent the maximum of the analytical uncertainty (0.06‰) and the standard deviation of
individual measurements in the month for each site. The colored lines represent the simulated zonal mean, while the
colored symbols represent the simulation sampled at the location of the GMD observations.

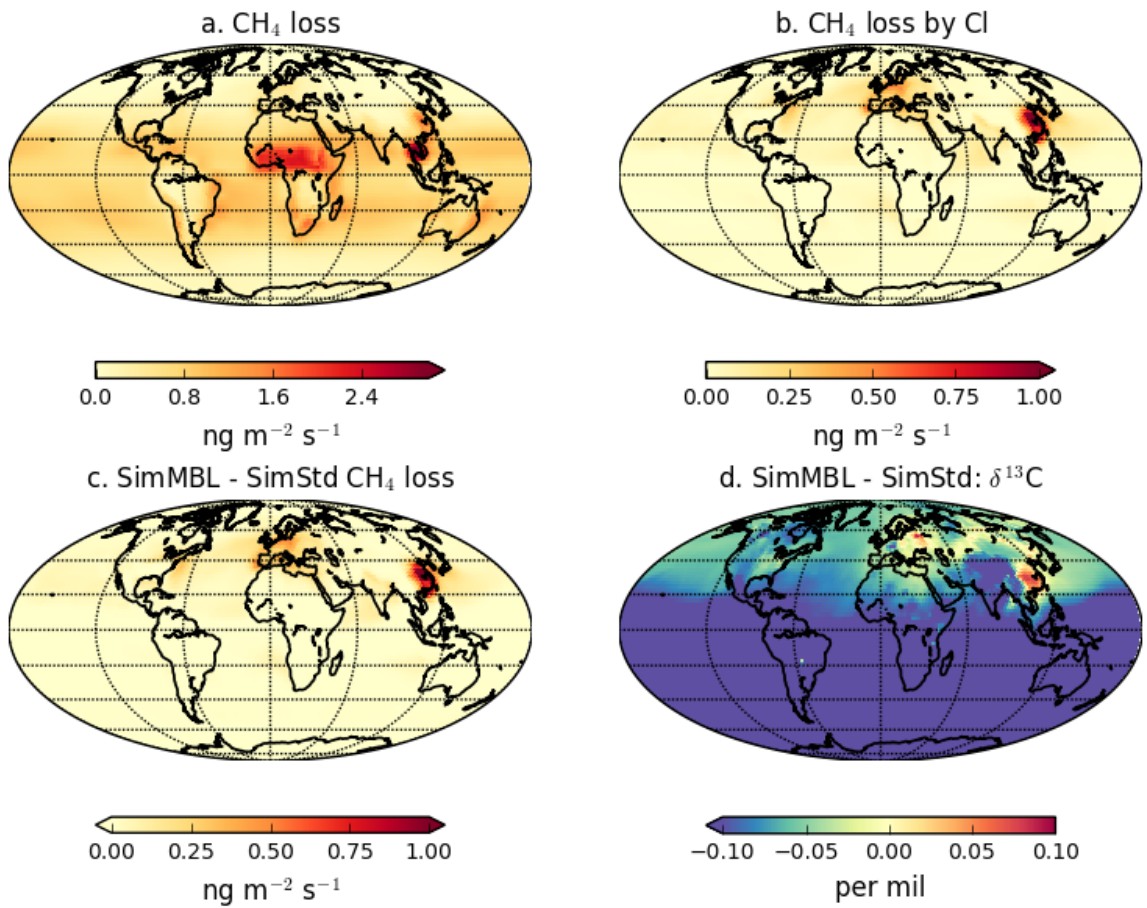

**Fig 8:** January a) $CH_4$ loss and b) $CH_4$ loss by Cl only in the SimTom simulation, as well as the difference in c)
$CH_4$ loss and d) $\delta^{13}$C-$CH_4$ between the SimTom and SimStd simulations.

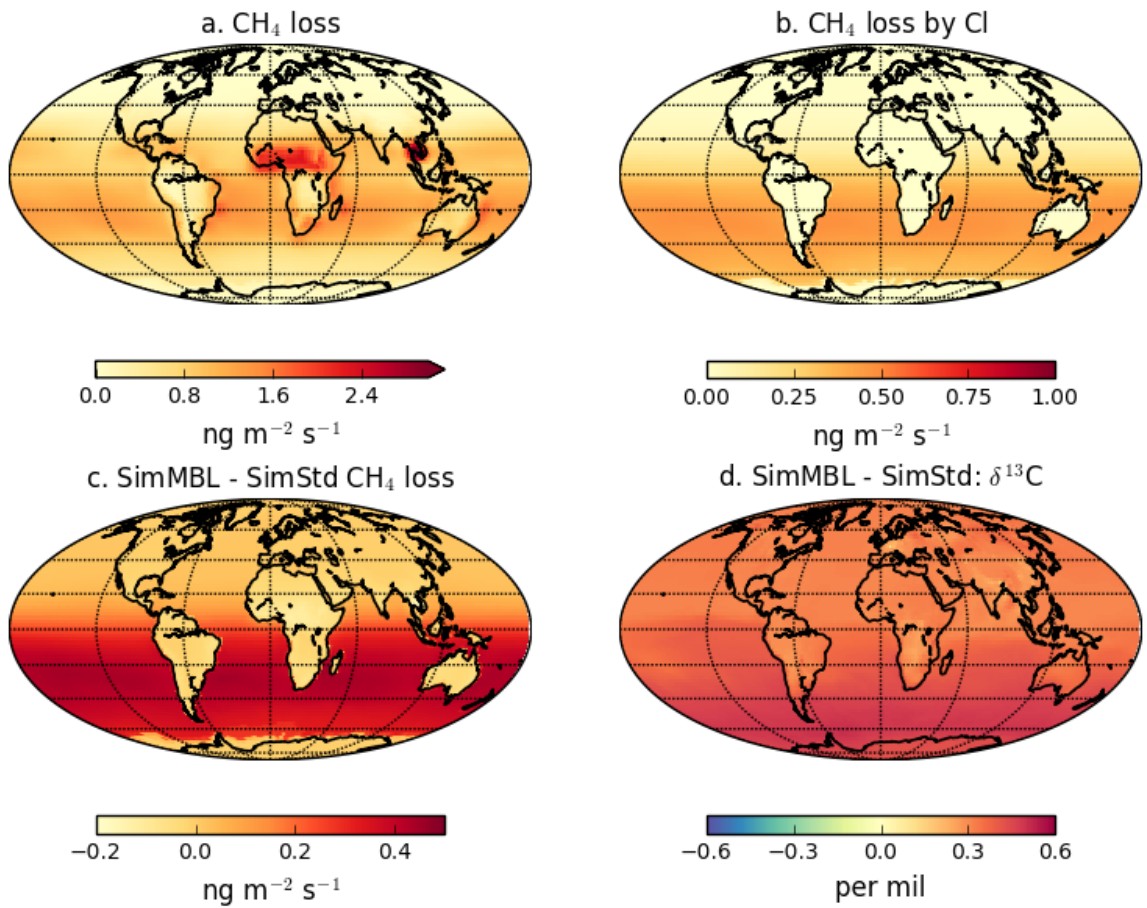

### a. CH₄ loss

ng m⁻² s⁻¹

### b. CH₄ loss by Cl

ng m⁻² s⁻¹

### c. SimMBL - SimStd CH₄ loss

ng m⁻² s⁻¹

### d. SimMBL - SimStd: $\delta^{13}C$

per mil

**Fig 9:** January a) CH₄ loss and b) CH₄ loss by Cl only in the SimMBL simulation, as well as the difference in c) CH₄ loss and d) $\delta^{13}C$-CH₄ between the SimMBL and SimStd simulations.

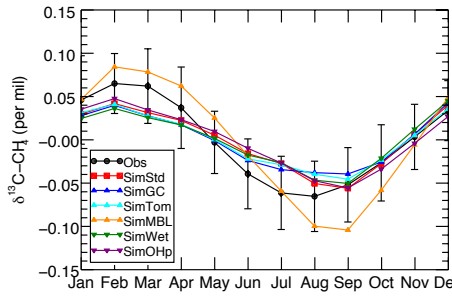

**Fig. 10:** The seasonal cycle of $\delta^{13}C$ of CH₄ at the SPO site with the annual mean removed averaged over 2002-2004 for the GMD observations (black), SimStd (red), SimGC (blue), SimTom (cyan), SimMBL (orange), SimWet (green), and SimOHp (purple). Errorbars represent the standard error, calculated as the maximum of the pooled standard deviation or the analytical uncertainty (0.06‰), divided by the square root of the number of years of observations.

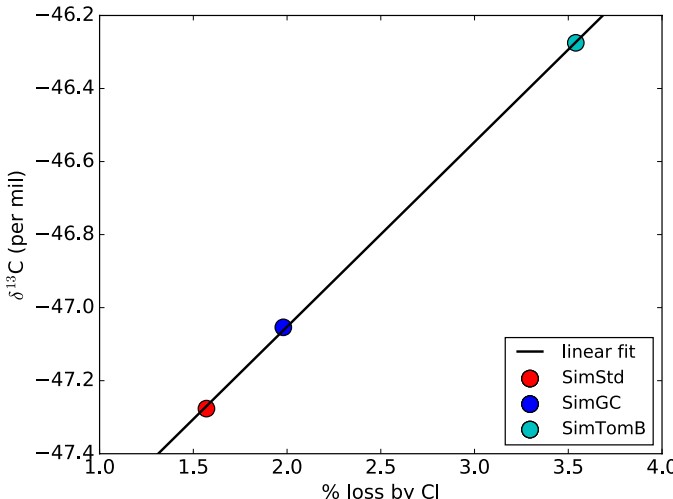

Fig. 11: Area-weighted global mean surface $\delta^{13}C$ for the SimStd (red), SimGC (blue) and SimTomB (cyan)
simulations in 2004 as a function of the percent of $CH_4$ loss occurring by reaction with Cl. The linear best-fit
line is shown in black.