# Peer review of "Strong Sensitivity of the Isotopic Composition of Methane to the Plausible Range of Tropospheric Chlorine"

_Atmospheric Chemistry and Physics, 2019_

## Short Comment (SC1) · 19 Dec 2019

In the study, OH concentrations were reduced proportionally to the proportion of the sink that the tropospheric Cl represented. However, in reality the relationship between OH and Cl concentrations and the lifetime and thus the $CH_4$ mole fraction is much more complex and non-linear, and depends also on the geographic and seasonal distribution of the OH, Cl and $CH_4$ fields. I believe the fact that the SimStd and SimGC simulations underestimate and overestimate, respectively, the observed $\delta^{13}CH_4$ in 2004, despite the fact that both include only a relatively small CH4 sink from Cl arise from this issue. This makes the scenarios less comparable. This is a forward simulation and as stated

by the authors, the isotopic composition is not in steady state, then why should the $CH_4$ be in steady state? I fear reducing OH may mask some of the impact of Cl on the $CH_4$ mole fraction. The fact that neither the $CH_4$ mole fraction nor the $\delta^{13}CH_4$ are in steady state could be dealt with by subtracting the measured value at SPO.

Furthermore, if one of the objectives of this study was to compare the modeled gradient with observations, why were the OH fields scaled to be higher in the Northern Hemisphere when observational evidence points towards a NH/SH ratio of $0.97 \pm 0.12$ (Patra et al., 2015)? In the same spirit, if the spatial variations of the $\delta^{13}CH_4$ signal of emissions has been proven to be important to reproduced the observed signals in the atmosphere (e.g. Schwietzke et al., 2016, Ganesan et al., 2018) why is the SimWet only a sensitivity case and not the standard?

I would like to make you aware of a possible conceptual error. Your VPDB value (0.0112372) is the old Craig (1957) value. This reference material has long been depleted. The actually valid number is 0.0111802 (recognized by IUPAC) from Zhang (1990). Please note that the measured reference in the Zhang (1990) paper was NBS 19, which defines the VPDB scale with $\delta^{13}$ = +1.95 per mill (Berglund and Wieser, 2011). Also, you may be treating the $CH_4$ mole fraction as $^{12}CH_4$. However, for the $CH_4$ mixing ratio the mass range covers all masses from 16 ($^{12}CH_4$) to 21 ($=^{13}CD_4$). Gas chromatography does not discriminate against higher isotopologues. Laser-based instruments, in contrast, do actually measure $^{12}CH_4$, but they are calibrated with gases whose composition was determined by gas chromatography. Thus the calibration gases must have a realistic isotopic composition to correct for this issue. To obtain, $^{12}CH_4$ you have to subtract the contribution of all other isotopologues from the mixing ratio.

I would like to point out some potential improvement for the transmission of information. I would find an analog to figure 5(c and d) for $\delta^{13}CH_4$ more useful than the maps. I would also recommend to use sine latitude in all the latitudinal profile plots to retain the proportion with respect to the Earth surface. The correlation coefficients of latitudinal

gradients of the different scenarios in figure 7 as well a comparison with their latitudinal derivative might also provide additional information. With this respect I believe figures figure S2 and S3 should be in the main text. Finally, a good validation for the different scenarios would be the reproduction of phase ellipses from Allan et al. (2001).

Finally some details in the text:

- In the methods section you describe simStd as

- No description of simWet in methods

- In the description of SimTom "very short lived substances" is repeated twice in the sentence

References

Allan, W., Manning, M. R., Lassey, K. R., Lowe, D. C., Gomez, A. J. (2001). Modeling the variation of $\delta$ 13C in atmospheric methane: Phase ellipses and the kinetic isotope effect. Global Biogeochemical Cycles, 15(2), 467. http://doi.org/10.1029/2000GB001282

Berglund, M., M. E. Wieser. Isotopic compositions of the elements 2009 (IUPAC Technical Report). Pure and Applied Chemistry 2011, 83, 397.

Craig, H., Isotopic Standards for Carbon and Oxygen and Correction Factors for Mass-Spectrometric Analysis of Carbon Dioxide. Geochimica Et Cosmochimica Acta 1957, 12, 133.

Ganesan, A. L., Stell, A. C., Gedney, N., Comyn-Platt, E., Hayman, G., Rigby, M., et al. (2018). Spatially Resolved Isotopic Source Signatures of Wetland Methane Emissions. Geophysical Research Letters, 45(8), 3737–3745. http://doi.org/10.1029/2008GB003299

[Figure]

Patra, P. K., Krol, M. C., Montzka, S. A., Arnold, T., Atlas, E. L., Lintner, B. R., et al. (2015). Observational evidence for interhemispheric hydroxyl-radical parity. Nature, 513(7517), 219–223. http://doi.org/10.1038/nature13721

Schwietzke, S., Sherwood, O. A., Bruhwiler, L. M. P., Miller, J. B., Etiope, G., Dlugokencky, E. J., et al. (2016). Upward revision of global fossil fuel methane emissions based on isotope database. Nature, 538(7623), 88–91. http://doi.org/10.1038/nature19797

Q. L. Zhang, T. L. Chang, W. J. Li. A Calibrated Measurement of the Atomic-Weight of Carbon. Chinese Science Bulletin 1990, 35, 290.

---

## Referee Comment (RC1) · Anonymous Referee #1 · 20 Jan 2020

This study addresses the interesting and important topic of methane oxidation by chlorine and the uncertainty it introduces in studies of global methane, in particular those that make use of d13C. The main conclusion of manuscript, which reads very well and is nicely concise and to the point, is that the d13C signature of atmospheric methane is sensitive to the treatment of Cl. To be honest, I was a bit disappointed by this – because in my perception that was quite clear already, and therefore doesn't bring much new. There are inherent limitations in doing forward simulations, but nevertheless. Having gone through the effort of fine-tuning model runs to be able to make realistic and meaningful comparisons with measurements, in my opinion some more in-depth analysis of the uncertainties concerning Cl should have been made to reach the level of impact of

the journal. Besides this I have a few methodological issues that require attention to make this work publishable.

GENERAL COMMENTS

I was surprised to read that despite the effort made by Schwietzke et al (2016), which is referenced in this paper, to revisit and update isotope signatures by systematically exploring what has been published in recent years, the outcome is practically unused here. It would be useful to know if it has any implications for the latitudinal and seasonal measurement constraints that are central to this study.

The authors rightly mention the limitation of using constant isotopic fractionation factors. The SimWet scenario is introduced to account to variations in isotopic signature for natural wetlands based on the work of Ganesan et al (2018). I was surprised to see an impact of just below 0.2 per mil on the north-south gradient, whereas looking it their figure 3 the impact could easily be 0.3 per mil (the impact is a bit larger along the Arctic cost where the measurement sites are located). 0.1 per mil is about the gap that remains with the measurements in Figure 7. The difference could be due to the size of the global wetland emission that is used, or how it is distributed latitudinally, which introduces an uncertainty that is worth considering.

The comparison between simulated and observed CH4 in Figure 2 is very convincing. However, limited information is provided to judge the d13C simulation and whether its trend is modelled realistically. The seasonal cycles in the supplement only show deviations from the mean, which are useful, but not sufficient to judge the overall performance. In my opinion the trend in d13C could provide important evidence about the role of Cl chemistry, as a sizeable source component should have been increasing over time – for which it remains the question whether its contribution to the d13C trend is confirmed by the measurements.

It is unclear why only measurements at the South Pole are used to assess the seasonal cycle of d13C. According to the work of Allen et al., the SimMBL scenario should

reproduce the seasonal observed at Baring Head. If so, the question is if the GEOS simulations confirm that this is the case, and what it means for the representativeness of either South Pole or Baring Head for the remote Southern Hemisphere. It is unclear why averages of multiple site are used for CH4, but not for d13C. The underestimated seasonal cycle at SPO for all the other simulations, with quite different representations of Cl, raises the question how sensitive the seasonal cycle amplitude really is to Cl. This depends not only on the size of the Cl sink, but also on its seasonal cycle, and whether or not it is in the right phase with what is observed. To properly judge this, a 'blank' scenario is missing without accounting for tropospheric chlorine. Do we actually need to account for the tropospheric CH4 sink due to Cl to be able to reproduce the CH4 and d13C measurements?

If the aim is to assess the sensitivity of global variations in d13C CH4 to the treatment of Cl, then an increased Cl sink should not be compensated by a decreasing KIE for OH. The argument that this is needed to keep the overall fractionation in agreement with observations is not so strong, given that earlier estimates of it are based mostly on measurements from Baring Head. The comparison at South Pole presented in this manuscript suggests that the fractionation may be different (depending on the corresponding seasonal cycle amplitude of CH4 itself, which is not shown in Figure 10). Unless the combined Cl and OH fractionation is really outside the range of a well-defined observational constraint on what it should be, it would be better not to change the treatment of OH.

SPECIFIC COMMENTS

Line 83: 'old bias' What could be the implication of this bias? If the age would be younger, would that increase the signature from stratospheric CH4 oxidation in the SH troposphere, enlarging the north-south gradient in d13C?

Line 98: As pointed out in the comment of Tonatiuh Nunez Ramirez, there is confusion in the literature on the value of Rstd. PDB is not used anymore. Instead, VPDB is the

standard with a Rstd = 0.01118. A good reference would be Zhang and Lee (1990).

Line 168: This suggests that the other Cl fields do not show significant shifts in the seasonal phasing of Cl between the hemispheres. If so, this would be an important point to provide further information on.

Line 177: But the uncertainty of the monthly mean is not the std of the individual measurements that are averaged.

Line 237: Looking at Fig. 6 I do not quite see the impact of the geologic source in northern Asia mentioned here. Given the modest emission from single geological formations, I wonder how the interhemispheric gradient can be so sensitive to it.

Line 250: Although the d13C simulations do show important differences when varying the treatment of Cl, I did not see a quantification of its significance for the global CH4 budget. This statement assumes some significant shift in sources in order to explain the measurements, depending on the treatment of Cl. Since this is not what is done in this study, I do not think it can formally be concluded from the results. Since the aim of the study is to assess the importance of the treatment of Cl for global CH4, I do think some kind of quantification of that importance is needed. However, the conclusion section doesn't provide a single number in support of a conclusion regarding the importance of Cl.

———————————————————

---

## Referee Comment (RC2) · Anonymous Referee #2 · 5 Feb 2020

This study investigates the use of different tropospheric chlorine sinks on the isotopic composition of methane. Through using a series of model runs, the authors test the impact of different Cl fields from the published literature on the isotopic composition of methane in the troposphere. As the authors highlight, because the reaction between CH4 and Cl is highly fractionating, a small change in tropospheric chlorine can have a substantial impact on the isotopic ratio of CH4. As such, the work serves as a reminder not to ignore the impacts of tropospheric chlorine on analyses of the methane carbon isotope ratio. In this regard the work has the potential to be highly relevant to many researchers in the field and sits well within the journal's scope. However, beyond this general point, I'm at a slight loss as to what has been learnt from this

study. The conclusions are entirely qualitative and merely serve to reiterate the point that an understanding of the chlorine sink is important when it comes to interpreting isotope ratios. I do not disagree with this point, but surely this work could aim higher by quantifying the impact on isotopic methane ratios (and source composition) of different assumptions regarding the chlorine sink. The paper does not want for brevity, so the inclusion of further analysis would not make it unduly long. I certainly think that there is merit in this work but it could do with a little more refining.

General comments:

Beyond the general point that the chlorine sink is important, the take-home message of the paper is rather vague. I think this is partially because of the confusing nature of the sensitivity experiments, where multiple variables are changed simultaneously (sink magnitude, sink distribution and fractionation rates) making it all but impossible to draw conclusions on what the key underlying factors are. For instance, although the SimStd and SimGC runs straddle the observations, is that due to the different magnitude of the sink or the different distribution? The SimTom simulation changes all variables at once, leading to an isotope ratio that at first appears conceptually wrong. Despite the larger Cl sink, the isotopic composition becomes more depleted in 13C! There is a throwaway line (P.6 L202) that the larger Cl sink is compensated by lower OH, but really this needs to be shown explicitly. Perhaps the point is that the Cl sink can't be too large, because the OH sink would have to be substantially smaller, making the simulated isotope ratio inconsistent with the observations? But, if so, then more work needs to be done to consider plausible changes to emissions totals so that one wouldn't have to change the OH sink to keep total CH4 consistent with the observations.

The results focus on the model outputs from a single year (2004), and to be more precise from 2 months (January and July) from this single year. However, I was not convinced of the reason why, and it would be helpful if more justification for this were provided. Perhaps I missed the point but why do flat methane concentrations simplify the analysis, when the focus is on the isotopic ratio? What does the isotopic composition trend look like during the same period(1990-2004), and is this well captured by the model (and sensitivity studies)? The authors note (p.6 L205) that since the isotopic composition is not in steady state in 2004, the results diverge with further years of simulation. So, does that mean the conclusion that SimStd and SimGC bracket the observations is only valid in 2004, and if so, what about in other years?

The conclusions seem to be mostly qualitative, raising the question of what the implications are for researchers wanting to use methane isotope data to constrain the evolution of different sources over time? Although it is stated that the choice of Cl field strongly impacts the CH4 source mixture that best fits the observations it would certainly help if this were spelled out more explicitly with some quantitative examples.

Specific comments:

p.3 L89-90 – Is the OH field seasonally varying but inter-annually constant? Please state.

p.3 L89-92 – Given the observational evidence that the NH/SH OH ratio is roughly uniform (e.g. Patra et al 2014) what is the justification for model output ratios with significant asymmetry? Indeed, the Strode et al (2015) paper that is cited seems to suggest that a ratio close to 1 also provides a better match to observations, so the use of a northern hemisphere OH field that is 20% higher is even more confusing.

P3. L106 – There is a reference missing for the fractionation value due to soil absorption. Also, how large is the soil sink as a fraction of total loss? How well is this constrained in the literature? Given the fractionation seems quite large, couldn't small errors in the soil absorption sink also impact on the isotopic ratio of atmospheric methane?

P3. L106 and Fig. 1 – I do not see the value of Figure 1 and any information it may show is largely duplicated in Figure 6. Perhaps it would do better in the supplement. The color gradient is also pretty weak, so if the figure is to be included it might be best

to reduce the scale so that differences are easier to determine.

P4. L133-134 - "We find the resulting emissions lead to a good simulation..." Please define "good", be it through RMSE, bias, correlation, etc.. Also please define NOAA GMD.

P4. L136-149 - What is the temporal variation in the Cl fields. Do they change inter-annually?

P4. L138-139 – For the avoidance of doubt, do all sensitivity studies use the same stratospheric loss fields? Please be explicit.

P4. L146-147 – Is the 2.5% loss the fraction in the TOMCAT model or in your model runs? I have assumed the latter but it is unclear.

P5. L154-155 – The Cl field from TOMCAT accounts for 2.5% of CH4 loss. But then the OH concentration is reduced by 2%, so does the Cl sink now account for more than 2.5% of CH4 loss?

P5. L154-155 – I appreciate the wish to keep total CH4 roughly the same but why do this by altering the OH loss rather than altering the total emissions? Is a 2% change in OH consistent with the oxidation of other species beyond methane such as CO or methyl chloroform? In other words, is this a plausible scenario? The same goes for the 4% reduction used in SimMBL. Wouldn't it be worth increasing the sources to compensate for the increase in sink instead?

P5. L155-156 – The fractionation value is increased in SimTom to avoid too much fractionation. But then the results show that there is too little. So is this change really justified? No evidence is shown for the case where the fractionation is kept the same to support this decision.

P5. L171-177 – I think it would be pertinent to include some mention of the distribution of observations and how many different locations there are measuring 13C in the observation network.

P5. L184 – Why choose to focus only on January and June? The answer may be obvious but it still needs to be mentioned.

P6. L194-196 –Perhaps it might help to give the latitude of this isotopically heavy region (∼60 N) to help guide the reader and link to Figure 7.

P6. L205-208 – Should the isotopic composition be in steady state in 2004? Although Fig. 2 shows the model and observed trends in bulk CH4, shouldn't the comparison of the modelled and observed isotopic ratios also be shown to assess the model performance? I.e Is it only in 2004 that the model isotope ratios are broadly in agreement with the observations?

P6. L215-219 – The point is made here that it is the distribution of the Cl sink that is key for the SimTom run, but I really think this needs to be made more prominent. The use of a different OH sink again complicates the issue as well, as the SH effects seem to be mostly a result of OH changes rather than Cl, so are you really analyzing the response to a change in the Cl sink or a change in the magnitude and fractionation of the OH sink?

P7. L221 – Given the reduced hemispheric OH difference improves the simulation again the question has to be asked why this wasn't used as the standard configuration?

P7. L236-237 – How large would the correction to the geologic source have to be to improve the inter-hemispheric gradient? It seems that the inter-hemispheric gradient is relatively unperturbed by the different Cl fields, and more likely a result of source differences or a change in the interhemispheric OH ratio. The simulation that best approximates the inter-hemispheric gradient is SimWet. Doesn't this point to accurate source signatures being of primary importance in determining this gradient, rather than the Cl field?

P7. L239-240 – Would it not be best to include SimWet description in section 2.2? Also it would be useful to give more details of this simulation. By how much does the

wetland source signature change versus the single value simulation?

P8. L265 - "We find that the NH Cl maximum acts to flatten the interhemispheric gradient...". But this isn't the Cl distribution alone that has this effect, as both the OH sink magnitude and fractionation changed. Based on the sensitivity cases shown one cannot conclude that this is down to Cl alone. The same applies for conclusions related to SimMBL.

P8. L271 - "...but combining it with the Cantrell et al (1990) value would lead to an over-estimate" Would it? This isn't shown. Changing the fractionating effect of OH between sensitivity cases without showing the evidence for why this is the most appropriate action seems odd.

P8. L272 – What about the uncertainty in the fractionating effect of other sinks such as soil absorption and Cl. Are they of a similar scale to the uncertainty in OH?

P8. L272-275 - "The choice of Cl field thus strongly impacts..." Yes, but by how much? Under these different assumptions about Cl how different would the source mixtures end up being?

Technical comments:

P. 3 L 83 – GMI is used before being defined

Fig. 4 – I appreciate the scales probably need to be different, but why do the panels use different color maps?

Fig 8 and 9 – Units omitted on panels b and c in both figures.

References: Patra, P., Krol, M., Montzka, S. et al. Observational evidence for interhemispheric hydroxyl-radical parity. Nature 513, 219–223 (2014). https://doi.org/10.1038/nature13721

---

## Author Comment (AC1) · 15 Apr 2020

**We thank Tonatiuh Guillermo Nuñez Ramirez for the thoughtful comment and respond to specific points below.**

In the study, OH concentrations were reduced proportionally to the proportion of the sink that the tropospheric Cl represented. However, in reality the relationship between OH and Cl concentrations and the lifetime and thus the CH4 mole fraction is much more complex and non-linear, and depends also on the geographic and seasonal distribution of the OH, Cl and CH4 fields. I believe the fact that the SimStd and SimGC simulations underestimate and overestimate, respectively, the observed δ 13CH4 in 2004, despite the fact that both include only a relatively small CH4 sink from Cl arise from this issue. This makes the scenarios less comparable.

**We do not alter the OH field between the SimStd and SimGC simulations, so differences in OH do not drive the differences between those simulations. In general, multiple factors including complex differences between the models' chemical mechanisms (not just Cl) drive OH differences between models. Since the purpose of this study is to quantify the effect of Cl on the isotopic composition of methane, we keep the scaling of OH for the SimTom and SimMBL simulations as simple as possible since it would otherwise be more difficult to disentangle differences due to Cl from differences due to OH distribution.**

This is a forward simulation and as stated by the authors, the isotopic composition is not in steady state, then why should the CH4 be in steady state? I fear reducing OH may mask some of the impact of Cl on the CH4 mole fraction. The fact that neither the CH4 mole fraction nor the δ 13CH4 are in steady state could be dealt with by subtracting the measured value at SPO.

**$CH_4$ in the real atmosphere is not in steady state for either total $CH_4$ or the isotopic ratio, but as described by Tans (1997), the isotopic ratio takes longer to adjust to a perturbation than total $CH_4$. The purpose of our statement on line 206, "the isotopic composition is not in steady state in 2004 and the results of the sensitivity simulations diverge further with additional years of simulation, with SimMBL becoming clearly inconsistent with observations", is to indicate that the temporal evolution of $\delta^{13}C$ in the SimMBL simulation does not match the temporal evolution of the observations.**

Furthermore, if one of the objectives of this study was to compare the modeled gradient with observations, why were the OH fields scaled to be higher in the Northern Hemisphere when observational evidence points towards a NH/SH ratio of $0.97 \pm 0.12$ (Patra et al., 2015)? In the same spirit, if the spatial variations of the δ13CH4 signal of emissions has been proven to be important to reproduced the observed signals in the atmosphere (e.g. Schwietzke et al., 2016, Ganesan et al., 2018) why is the SimWet only a sensitivity case and not the standard?

**We altered the NH/SH ratio of the OH field to better align with what is typically simulated by global chemistry models, since we are using Cl fields from global chemistry models. Since the isotopic methane budget is under-constrained, there are many possible combinations of sources, source isotopic signatures, sink distributions, etc. that we could have chosen for our standard simulation, and some combinations would likely match observations better than SimStd. However, our goal was not to optimize the fit to observations, but rather to quantify the strong sensitivity of $\delta^{13}C$ to the range of Cl**

**concentrations reported in the literature. The finding of strong sensitivity to Cl does not depend on the specific NH/SH OH ratio or source signatures used in SimStd.**

I would like to make you aware of a possible conceptual error. Your VPDB value (0.0112372) is the old Craig (1957) value. This reference material has long been depleted. The actually valid number is 0.0111802 (recognized by IUPAC) from Zhang (1990). Please note that the measured reference in the Zhang (1990) paper was NBS 19, which defines the VPDB scale with δ 13 = +1.95 per mill (Berglund and Wieser, 2011).

**It is true that we, like many others in the literature, used the old value both for partitioning the total methane source into 12C and 13C components, and for converting our simulated 13C and 12C concentrations to $\delta^{13}$C. We felt that using the old value for the source partitioning was most consistent with the isotopic source signatures in the literature, such as the compilations by Houweling et al (2000) and Lassey et al (2007), which both cite the Craig (1957) number. We then used the same number for calculating $\delta^{13}$C for consistency. However, we tested the impact of this choice by conducting an additional simulation identical to SimStd using the Zhang (1990) value for both the source partitioning and the calculation of $\delta^{13}$C. The new simulation gives very similar results to SimStd, demonstrating that our results are robust to the choice of VPDB as long as the same value used to calculate the source partitioning is used to calculate $\delta^{13}$C. We elaborate on this important in the Model Description section of our revised manuscript.**

Also, you may be treating the CH4 mole fraction as 12CH4. However, for the CH4 mixing ratio the mass range covers all masses from 16 (12CH4) to 21 (=13CD4). Gas chromatography does not discriminate against higher isotopologues. Laser-based instruments, in contrast, do actually measure 12CH4, but they are calibrated with gases whose composition was determined by gas chromatography. Thus the calibration gases must have a realistic isotopic composition to correct for this issue. To obtain, 12CH4 you have to subtract the contribution of all other isotopologues from the mixing ratio.

**We treat CH$_4$ mol fraction as the sum of $^{12}$CH$_4$ and $^{13}$CH$_4$. Our simulation only includes $^{13}$CH$_4$ and $^{12}$CH$_4$, as there are currently not sufficient observations available globally to constrain simulations of e.g. $^{13}$CD$_4$. Given the very minor relative abundance of the other isotopologues, we believe this error is very small compared to other uncertainties in the methane budget.**

I would like to point out some potential improvement for the transmission of information. I would find an analog to figure 5(c and d) for δ 13CH4 more useful than the maps.

**Fig. 7 shows the isotopic information in a similar way (as a function of latitude). We also add an additional figure showing time series of the isotopic values at individual stations.**

I would also recommend to use sine latitude in all the latitudinal profile plots to retain the proportion with respect to the Earth surface. The correlation coefficients of latitudinal gradients of the different scenarios in figure 7 as well a comparison with their latitudinal derivative might also provide additional information.

**The spatial correlation is provided in Section 3.2.**

With this respect I believe figures figure S2 and S3 should be in the main text. Finally, a good validation for the different scenarios would be the reproduction of phase ellipses from Allan et al. (2001).

**Since we added additional figures to the main text in the revised manuscript, we prefer to keep S2 and S3 in the Supplemental section. We experimented with including the phase ellipses, but found that they were noisy and provided little additional insight compared to what was already shown in the season cycle plots.**

Finally some details in the text: • In the methods section you describe simStd as • No description of simWet in methods • In the description of SimTom "very short lived substances" is repeated twice in the sentence

**We added a description of SimWet to the methods section.**

**References:**

**Craig, H.: Isotopic Standards for Carbon and Oxygen and Correction Factors for Mass-Spectrometric Analysis OF Carbon Dioxide, Geochimica Et Cosmochimica Acta, 12, 133-149, 10.1016/0016-7037(57)90024-8, 1957.**

**Houweling S, Dentener F, and Lelieveld J: Simulation of preindustrial atmospheric methane to constrain the global source strength of natural wetlands, J. Geophys. Res., 105(D13), doi:10.1029/2000JD900193, 2000.**

**Lassey KR, Etheridge DM, Lowe DC, Smith AM, Ferretti DF: Centennial evolution of the atmospheric methane budget: what do the carbon isotopes tell us? Atmos Chem Phys 7(8): 2119-2139. doi:10.5194/acp-7-2119-2007, 2007.**

**Tans P. 1997. A note on isotopic ratios and the global atmospheric methane budget. *Global Biogeochemical Cycles* 11(1): 77-81. doi:10.1029/96GB03940.**

**Zhang, Q.-L. and Li, W.-J.: A Calibrated Measurement of the Atomic Weight of Carbon, Chinese Science Bulletin 35, 290, doi: 10.1360/sb1990-35-4-290, 1990.**

---

## Author Comment (AC2) · 15 Apr 2020

**We thank Referee #1 for the thoughtful comments and respond to individual comments below. Our responses are in bold.**

Anonymous Referee #1

This study addresses the interesting and important topic of methane oxidation by chlorine and the uncertainty it introduces in studies of global methane, in particular those that make use of d13C. The main conclusion of manuscript, which reads very well and is nicely concise and to the point, is that the d13C signature of atmospheric methane is sensitive to the treatment of Cl. To be honest, I was a bit disappointed by this – because in my perception that was quite clear already, and therefore doesn't bring much new. There are inherent limitations in doing forward simulations, but nevertheless. Having gone through the effort of fine-tuning model runs to be able to make realistic and meaningful comparisons with measurements, in my opinion some more in-depth analysis of the uncertainties concerning Cl should have been made to reach the level of impact of the journal. Besides this I have a few methodological issues that require attention to make this work publishable.

**We now clarify in the abstract and introduction that our study shows how inter-model diversity in the Cl simulated by state-of-the art global models, not just uncertainty in Cl in general, impacts $\delta^{13}$C. We believe this is a novel analysis.**

**We now state in the abstract: "Global model simulations of halogen chemistry differ strongly from one another in terms of both the magnitude of tropospheric Cl and its geographic distribution. This study explores the impact of the inter-model diversity in Cl fields on the simulated d$^{13}$C of CH$_4$." And "Consequently, it is possible to achieve a good representation of total CH$_4$ using widely different Cl concentrations, but the partitioning of CH$_4$ loss between the OH and Cl reactions leads to strong differences in isotopic composition depending on which model's Cl field is used." We clarify in the introduction: "Here, we investigate the sensitivity of $\delta^{13}$C of CH$_4$ to inter-model diversity in chlorine concentrations to better quantify how much uncertainty in the interpretation of $\delta^{13}$C is imposed by the uncertainty in Cl."**

**We also added an additional section (Section 3.3) quantifying the relationship between the percent of methane loss from Cl and the surface $\delta^{13}$C. This addition provides a useful quantitative measure of the impact of Cl on the isotopic budget.**

GENERAL COMMENTS
I was surprised to read that despite the effort made by Schwietzke et al (2016), which is referenced in this paper, to revisit and update isotope signatures by systematically exploring what has been published in recent years, the outcome is practically unused here. It would be useful to know if it has any implications for the latitudinal and seasonal measurement constraints that are central to this study.

**Latitudinal variations in source signatures can indeed influence the isotopic distribution, as we acknowledge in our conclusions when we state "the interhemispheric gradient is also**

**influenced by spatial variation in the isotopic signatures of the sources, complicating this issue." We also mention in Section 3.4 that "Including spatially-varying isotopic signature for other sources as well could further modify the simulated interhemispheric gradient, potentially correcting some of the flat gradient of e.g. the SimTom simulation". We do incorporate some of the spatial variation in source signatures by separating biomass burning into C3 and C4 fractions with different isotopic signatures, and our SimWet simulation also incorporates the spatial variability of wetland emissions. Incorporation of even more detailed or updated isotopic source data is a useful direction for future work, but beyond the scope of the current study.**

The authors rightly mention the limitation of using constant isotopic fractionation factors. The SimWet scenario is introduced to account to variations in isotopic signature for natural wetlands based on the work of Ganesan et al (2018). I was surprised to see an impact of just below 0.2 per mil on the north-south gradient, whereas looking it their figure 3 the impact could easily be 0.3 per mil (the impact is a bit larger along the Arctic cost where the measurement sites are located). 0.1 per mil is about the gap that remains with the measurements in Figure 7. The difference could be due to the size of the global wetland emission that is used, or how it is distributed latitudinally, which introduces an uncertainty that is worth considering.

**This is an interesting point. Wetland emissions in our simulation are smaller than those of Ganesan et al (2018), and the strength of other sources differs as well, so we do not expect the impact to be identical. We add the following statement to section 3.4 of our revised manuscript: "The size of the effect of including spatially varying ratios in wetland emissions depends on the strength of the wetland emissions as well as the other sources."**

The comparison between simulated and observed CH4 in Figure 2 is very convincing. However, limited information is provided to judge the d13C simulation and whether its trend is modelled realistically. The seasonal cycles in the supplement only show deviations from the mean, which are useful, but not sufficient to judge the overall performance. In my opinion the trend in d13C could provide important evidence about the role of Cl chemistry, as a sizeable source component should have been increasing over time – for which it remains the question whether its contribution to the d13C trend is confirmed by the measurements.

**We add an additional figure, included below, showing the simulated and observed $\delta^{13}$C for 1998-2004 for 6 GMD sites with records extending back to 1998, the year data becomes available. This figure shows that the differences between sensitivity simulations are large compared to the trend in the observations over this period, so our conclusions are not specific to 2004. Investigating the trend in the isotopic observations is beyond the scope of our study since the portion of our simulation covered by the GMD observations is short for trend analysis. We add the following text to Section 3.2:**

**"Figure 6 shows the timeseries of observed and simulated $\delta^{13}$C for 1998-2004 at the 6 GMD sites with $\delta^{13}$C records covering this time period. We begin the figure at 1998 rather than 1990 due to the lack of data availability in the earlier years. The standard and sensitivity**

simulations overestimate $\delta^{13}C$ at the northernmost station, BRW. The observations at the other stations lie within the range of simulations, with most simulations underestimating the observations at the South Pole. The differences between the different sensitivity simulations are large compared to the interannual variability in both observed and simulated $\delta^{13}C$. We focus our subsequent analysis focuses on a single year, 2004."

[Figure]

It is unclear why only measurements at the South Pole are used to assess the seasonal cycle of d13C. According to the work of Allen et al., the SimMBL scenario should reproduce the seasonal observed at Baring Head. If so, the question is if the GEOS simulations confirm that this is the case, and what it means for the representativeness of either South Pole or Baring Head for the remote Southern Hemisphere.

**We show the seasonal cycle at other stations in Supplemental Figure S5. The seasonal cycle is also apparent in our new figure showing the 1998-2004 time series of $\delta^{13}C$. GMD data is not available at Baring Head for the years of our simulation. However, we discuss another southern hemisphere site, CGO, which shows a similar result to the South Pole.**

It is unclear why averages of multiple site are used for CH4, but not for d13C.

**We used averages for CH₄ to avoid excessive numbers of plots, particularly since total CH₄ is not our main focus. There are less sites with isotopic data, and that is our main focus, so we did not feel it necessary to take averages.**

The underestimated seasonal cycle at SPO for all the other simulations, with quite different representations of Cl, raises the question how sensitive the seasonal cycle amplitude really is to Cl. This depends not only on the size of the Cl sink, but also on its seasonal cycle, and whether or not it is in the right phase with what is observed. To properly judge this, a 'blank' scenario is missing without accounting for tropospheric chlorine. Do we actually need to account for the tropospheric CH4 sink due to Cl to be able to reproduce the CH4 and d13C measurements?

**It is difficult to definitively answer this question since the seasonal cycle is also influenced by the seasonal cycle and isotopic signatures of the sources, as we show with the SimWet simulation. However, we do show the potential for a strong MBL Cl source to alter the amplitude of the seasonal cycle in the southern hemisphere with minimal impact on northern hemisphere sites. We add the following discussion to Section 3.2:**

**"However, since the seasonal cycle amplitude at SPO lies in between SimMBL and the other simulations, it is possible that at an MBL Cl source similar to that of SimMBL but with a smaller average value could reproduce the amplitude well."**

If the aim is to assess the sensitivity of global variations in d13C CH4 to the treatment of Cl, then an increased Cl sink should not be compensated by a decreasing KIE for OH. The argument that this is needed to keep the overall fractionation in agreement with observations is not so strong, given that earlier estimates of it are based mostly on measurements from Baring Head. The comparison at South Pole presented in this manuscript suggests that the fractionation may be different (depending on the corresponding seasonal cycle amplitude of CH4 itself, which is not shown in Figure 10). Unless the combined Cl and OH fractionation is really outside the range of a welldefined observational constraint on what it should be, it would be better not to change the treatment of OH.

**The combined Cl and OH fractionation is indeed well outside the observed constraint. However, we recognize that changing the KIE of OH complicates our analysis, so we added an additional sensitivity study, SimTomB, that uses the Cl field of the SimTom simulation but keeps the same KIE of OH as the standard simulation. We use this simulation in the new figure 11, shown below, and add a new Section 3.3 that quantifies the effect of changing Cl when OH fractionation is held constant.**

[Figure]

SPECIFIC COMMENTS
Line 83: 'old bias' What could be the implication of this bias? If the age would be younger, would that increase the signature from stratospheric CH4 oxidation in the SH troposphere, enlarging the north-south gradient in d13C?

**The old bias here relates to the time since the air was in contact with the northern hemisphere midlatitudes, so it is also affected by tropospheric transport. We expect the impact on isotopic composition to depend on the details of the transport bias, which is beyond the scope of this study. However, the bias is small enough that we expect it to be a small uncertainty compared to the uncertainties due to methane's sources and sinks, and their isotopic signatures.**

Line 98: As pointed out in the comment of Tonatiuh Nunez Ramirez, there is confusion in the literature on the value of Rstd. PDB is not used anymore. Instead, VPDB is the C3
standard with a Rstd = 0.01118. A good reference would be Zhang and Lee (1990).

**It is true that we, like many others in the literature, used the old value both for partitioning the total methane source into 12C and 13C components, and for converting our simulated 13C and 12C concentrations to $\delta^{13}$C. We felt that using the old value for the source partitioning was most consistent with the isotopic source signatures in the literature, such as the compilations by Houweling et al (2000) and Lassey et al (2007), which both cite the Craig (1957) number. We then used the same number for calculating $\delta^{13}$C for consistency. However, we tested the impact of this choice by conducting an additional simulation identical to SimStd using the Zhang (1990) value for both the source partitioning and the calculation of $\delta^{13}$C. The new simulation gives very similar results to SimStd, demonstrating that our results are robust to the choice of VPDB as long as the same value used to calculate the source partitioning is used to calculate $\delta^{13}$C. We elaborate on this important in the Model Description section of our revised manuscript:**

**"We partition each emission source into $^{12}CH_4$ and $^{13}CH_4$ emissions according to a source-specific $\delta^{13}C$ value from the literature, provided in Table 1. We use the Craig (1957) $R_{std}$ value to partition the sources since it is cited in the literature used in Table 1 (Houweling et al, 2000; Lassey, 2007), and so for consistency we use the same value in equation 1 to calculate the simulated $d^{13}C$ of the $CH_4$ concentrations. We note, however, that the GMD observations now use a slightly different standard, the VPDB value of 0.011183 (Zhang and Li, 1990). A sensitivity study (not shown) confirms that the choice Rstd has little effect on our results as long as the same value is used for the source partitioning as for the calculation of $\delta^{13}C$-$CH_4$ from simulated [$^{13}CH_4$] and [$^{12}CH_4$]."**

Line 168: This suggests that the other Cl fields do not show significant shifts in the seasonal phasing of Cl between the hemispheres. If so, this would be an important point to provide further information on.

**This is a good suggestion. We add a supplemental figure showing the seasonal cycle by latitude of the four surface Cl fields. We add the following description to section 2.2: "SimStd and SimGC have more modest seasonal shifts, while Cl in SimTom remains concentrated in the northern hemisphere throughout the year (Fig. S3)."**

Line 177: But the uncertainty of the monthly mean is not the std of the individual measurements that are averaged.

**We felt it was more conservative to use the standard deviation rather than the standard error of the measurements within the month. However, when we average multiple years of data (for the seasonal cycle figures), we then use the pooled variance to calculate the standard error across years. We now clarify this in section 2.3 by adding: "When multiple years are observations are averaged together, we use the pooled variance to calculate the standard error, thus reducing the error based on the number of years."**

Line 237: Looking at Fig. 6 I do not quite see the impact of the geologic source in northern Asia mentioned here. Given the modest emission from single geological formations, I wonder how the interhemispheric gradient can be so sensitive to it.

**We clarified in the text that it is in "northern Eurasia (around 60°N)". Figure S1 shows that this source is quite large in the model.**

Line 250: Although the d13C simulations do show important differences when varying the treatment of Cl, I did not see a quantification of its significance for the global CH4 budget. This statement assumes some significant shift in sources in order to explain the measurements, depending on the treatment of Cl. Since this is not what is done in this study, I do not think it can formally be concluded from the results. Since the aim of the study is to assess the importance of the treatment of Cl for global CH4, I do think some kind of quantification of that importance is needed. However, the conclusion section doesn't provide a single number in support of a conclusion regarding the importance of Cl.

**We added an additional figure and an additional Section 3.3 dedicated to quantifying the link between CH₄ oxidation by Cl and the surface δ¹³CH₄. We use this analysis to add the following quantification of the Cl effect to the conclusions:**
**"Each percent increase in the amount of CH₄ loss occurring by reaction with Cl increases global mean surface δ¹³C of CH₄ by approximately 0.5‰. This relationship can be used to estimate the impact on methane's isotopic values from future model simulations of Cl."**

**We thank Referee #2 for the thoughtful comments and respond to individual comments below.**

Anonymous Referee #2

This study investigates the use of different tropospheric chlorine sinks on the isotopic composition of methane. Through using a series of model runs, the authors test the impact of different Cl fields from the published literature on the isotopic composition of methane in the troposphere. As the authors highlight, because the reaction between CH4 and Cl is highly fractionating, a small change in tropospheric chlorine can have a substantial impact on the isotopic ratio of CH4. As such, the work serves as a reminder not to ignore the impacts of tropospheric chlorine on analyses of the methane carbon isotope ratio. In this regard the work has the potential to be highly relevant to many researchers in the field and sits well within the journal's scope. However, beyond this general point, I'm at a slight loss as to what has been learnt from this study. The conclusions are entirely qualitative and merely serve to reiterate the point that an understanding of the chlorine sink is important when it comes to interpreting isotope ratios. I do not disagree with this point, but surely this work could aim higher by quantifying the impact on isotopic methane ratios (and source composition) of different assumptions regarding the chlorine sink. The paper does not want for brevity, so the inclusion of further analysis would not make it unduly long. I certainly think that there is merit in this work but it could do with a little more refining.

**We followed the reviewers' suggestion to make the paper more quantitative by adding an additional analysis, described in Section 3.3. This analysis quantifies the relationship between the fraction of methane oxidized by Cl and the response of surface δ¹³C. We believe that including this single number quantifying the impact of Cl enhances the take-home message of the paper.**

General comments:

Beyond the general point that the chlorine sink is important, the take-home message of the paper is rather vague. I think this is partially because of the confusing nature of the sensitivity experiments, where multiple variables are changed simultaneously (sink magnitude, sink distribution and fractionation rates) making it all but impossible to draw conclusions on what the key underlying factors are. For instance, although the SimStd

and SimGC runs straddle the observations, is that due to the different magnitude of the sink or the different distribution? The SimTom simulation changes all variables at once, leading to an isotope ratio that at first appears conceptually wrong. Despite the larger Cl sink, the isotopic composition becomes more depleted in 13C! There is a throwaway line (P.6 L202) that the larger Cl sink is compensated by lower OH, but really this needs to be shown explicitly. Perhaps the point is that the Cl sink can't be too large, because the OH sink would have to be substantially smaller, making the simulated isotope ratio inconsistent with the observations? But, if so, then more work needs to be done to consider plausible changes to emissions totals so that one wouldn't have to change the OH sink to keep total CH4 consistent with the observations.

**A major point of our study was to quantify the effect of the inter-model diversity in Cl fields produced by the current generation of chemistry climate models on methane's isotopic composition. We now clarify this purpose in the introduction and abstract. This inter-model diversity includes differences in the distribution as well as the magnitude of tropospheric Cl. These multiple differences do complicate our analysis as the reviewer points out, but we feel that including all aspects of the model diversity makes the study more applicable to understanding the impact of model differences.**

**While adjusting the OH fractionation was necessary to maintain consistency with the isotopic observations, we recognize that changing multiple variables at once makes quantifying the impact of Cl difficult. Consequently, we add an additional sensitivity simulation, SimTomB, that uses the same Cl as SimTom but keeps the same OH and OH fractionation as the SimStd and SimGC simulations. We then use these 3 simulations together in our new Section 3.3 and a new figure to quantify the impact of Cl. Since the Cl fields differ in distribution as well as magnitude between the 3 simulations, we use the total fraction of CH$_4$ lost through reaction with Cl as an integrated measure of these effects. This fraction is reported by other modeling studies, so we expect presenting our results this way will be useful.**

The results focus on the model outputs from a single year (2004), and to be more precise from 2 months (January and July) from this single year. However, I was not convinced of the reason why, and it would be helpful if more justification for this were provided. Perhaps I missed the point but why do flat methane concentrations simplify the analysis, when the focus is on the isotopic ratio? What does the isotopic composition trend look like during the same period(1990-2004), and is this well captured by the model (and sensitivity studies)? The authors note (p.6 L205) that since the isotopic composition is not in steady state in 2004, the results diverge with further years of simulation. So, does that mean the conclusion that SimStd and SimGC bracket the observations is only valid in 2004, and if so, what about in other years?

**We add an additional figure showing the simulated and observed $\delta^{13}$C for 1998-2004 for 6 GMD sites with records extending back to 1998, the year data becomes available. This figure shows that the differences between sensitivity simulations are large compared to the trend in the observations over this period, so our conclusions are not specific to 2004. We add the following text to Section 3.2:**

**"Figure 6 shows the timeseries of observed and simulated $\delta^{13}C$ for 1998-2004 at the 6 GMD sites with $\delta^{13}C$ records covering this time period. We begin the figure at 1998 rather than 1990 due to the lack of data availability in the earlier years. The standard and sensitivity simulations overestimate $\delta^{13}C$ at the northernmost station, BRW. The observations at the other stations lie within the range of simulations, with most simulations underestimating the observations at the south pole. The differences between the different sensitivity simulations are large compared to the interannual variability in both observed and simulated $\delta^{13}C$. We focus our subsequent analysis on a single year, 2004."**

The conclusions seem to be mostly qualitative, raising the question of what the implications are for researchers wanting to use methane isotope data to constrain the evolution of different sources over time? Although it is stated that the choice of Cl field strongly impacts the CH4 source mixture that best fits the observations it would certainly help if this were spelled out more explicitly with some quantitative examples.

**As we describe above, we now have a quantitative description of the impact Cl oxidation that can be used to estimate the impact on $\delta^{13}C$ from future model simulations of Cl. We added this to the conclusions:**
**"Each percent increase in the amount of CH$_4$ loss occurring by reaction with Cl increases global mean surface $\delta^{13}C$ of CH$_4$ by approximately 5‰. This relationship can be used to estimate the impact on methane's isotopic values from future model simulations of Cl."**

Specific comments:
p.3 L89-90 – Is the OH field seasonally varying but inter-annually constant? Please state.

**Yes, we add this to Section 2.1:**
**"The OH field varies monthly but repeats every year."**

p.3 L89-92 – Given the observational evidence that the NH/SH OH ratio is roughly uniform (e.g. Patra et al 2014) what is the justification for model output ratios with significant asymmetry? Indeed, the Strode et al (2015) paper that is cited seems to suggest that a ratio close to 1 also provides a better match to observations, so the use of a northern hemisphere OH field that is 20% higher is even more confusing.

**Since one of our goals is to understand the impacts of model diversity, we thought it best to use an OH distribution similar to what many CCMs, including GEOS when run as a CCM, produce. We now explain this in Section 2.1:**
**"This modification is designed to make our results more applicable to understanding the impacts inter-model differences in Cl, since it makes our OH distribution more consistent by that produced by many CCMs. The OH field varies monthly but repeats every year."**

P3. L106 – There is a reference missing for the fractionation value due to soil absorption. Also, how large is the soil sink as a fraction of total loss? How well is

this constrained in the literature? Given the fractionation seems quite large, couldn't small errors in the soil absorption sink also impact on the isotopic ratio of atmospheric methane?

**We added the reference to Tyler et al (1994). The soil sink is an additional source of uncertainty, and we now mention this in the conclusions:**
**"However, the interhemispheric gradient is also influenced by spatial variation in the isotopic signatures of the sources and uncertainties in the soil sink, complicating this issue."**

P3. L106 and Fig. 1 – I do not see the value of Figure 1 and any information it may show is largely duplicated in Figure 6. Perhaps it would do better in the supplement. The color gradient is also pretty weak, so if the figure is to be included it might be best to reduce the scale so that differences are easier to determine.

**We moved the figure to the supplement.**

P4. L133-134 - "We find the resulting emissions lead to a good simulation..." Please define "good", be it through RMSE, bias, correlation, etc..

**We add that "The simulation has only a 0.1% mean bias compared to the observations for 2004." Other statistics including correlation are provided in Section 3.1.**

Also please define NOAA GMD.

**Done**

P4. L136-149 - What is the temporal variation in the Cl fields. Do they change interannually?

**The Cl fields vary monthly but repeat from year to year. We added an additional supplemental figure showing the monthly variations of surface Cl. We added the following clarification to Section 2.2:**
**"SimStd and SimGC have more modest seasonal shifts, while Cl in SimTom remains concentrated in the northern hemisphere throughout the year (Fig. S3). All simulations repeat the same Cl field from year to year."**

P4. L138-139 – For the avoidance of doubt, do all sensitivity studies use the same stratospheric loss fields? Please be explicit.

**They are the same above 56 hPa. We now clarify in section 2.2:**
**"We also conduct several sensitivity simulations in which we alter the tropospheric and lower stratospheric Cl fields (Table 2). Cl is not altered above 56 hPa."**

P4. L146-147 – Is the 2.5% loss the fraction in the TOMCAT model or in your model runs? I have assumed the latter but it is unclear.

**We added "in our simulation" to clarify that it is in our model run.**

P5. L154-155 – The Cl field from TOMCAT accounts for 2.5% of CH4 loss. But then the OH concentration is reduced by 2%, so does the Cl sink now account for more than 2.5% of CH4 loss?

**No, the OH adjustment was already included.**

P5. L154-155 – I appreciate the wish to keep total CH4 roughly the same but why do this by altering the OH loss rather than altering the total emissions? Is a 2% change in OH consistent with the oxidation of other species beyond methane such as CO or methyl chloroform? In other words, is this a plausible scenario? The same goes for the 4% reduction used in SimMBL. Wouldn't it be worth increasing the sources to compensate for the increase in sink instead?

**These changes in OH are well within the uncertainty of OH derived from either methyl chloroform inversions or global models. We add this to section 2.2:**
**"These changes are small compared to the uncertainty in global OH."**

**We could have increased the sources instead, but this would lead to different emissions in different sensitivity runs, which would also complicate the analysis.**

P5. L155-156 – The fractionation value is increased in SimTom to avoid too much fractionation. But then the results show that there is too little. So is this change really justified? No evidence is shown for the case where the fractionation is kept the same to support this decision.

**We now have an additional sensitivity simulation, as noted above, that parallels SimTom but with the fractionation kept the same. This simulation quickly diverges from the observations.**

P5. L171-177 – I think it would be pertinent to include some mention of the distribution of observations and how many different locations there are measuring 13C in the observation network.

**We add the following information to Section 2.3:**
**"The GMD observations are located at remote sites, shown in Fig. 4 for CH$_4$ in 2004. Measurements of $\delta^{13}$C of CH$_4$ are available at a subset of the sites, shown in Fig. 5."**

P5. L184 – Why choose to focus only on January and June? The answer may be obvious but it still needs to be mentioned.

**We now clarify in Section 3.1:**
**"We focus on these two months to represent the seasonal differences."**

P6. L194-196 –Perhaps it might help to give the latitude of this isotopically heavy region

(_60 N) to help guide the reader and link to Figure 7.

**We now say:**
**"the isotopically heavy region in northern Eurasia (around 60°N)"**

P6. L205-208 – Should the isotopic composition be in steady state in 2004? Although
Fig. 2 shows the model and observed trends in bulk CH4, shouldn't the comparison of
the modelled and observed isotopic ratios also be shown to assess the model performance?
I.e Is it only in 2004 that the model isotope ratios are broadly in agreement
with the observations?

**As discussed above, we added a new figure 6 that shows a comparison for more years and**
**demonstrates that the results are not specific to 2004.**

P6. L215-219 – The point is made here that it is the distribution of the Cl sink that is
key for the SimTom run, but I really think this needs to be made more prominent. The
use of a different OH sink again complicates the issue as well, as the SH effects seem
to be mostly a result of OH changes rather than Cl, so are you really analyzing the
response to a change in the Cl sink or a change in the magnitude and fractionation of
the OH sink?

**Given the way the experiments are designed, we cannot completely isolate these effects. We**
**are showing the combined effects of assuming oxidation by Cl versus OH within the**
**observational constraints on the total provided by the observations. To clarify that**
**multiple effects are included, we now state in Section 3.2:**
**"The differences between simulations reflect differences in the locations where CH$_4$**
**oxidation occurs and the amount and location of isotopic fractionation due to Cl versus**
**OH".**

P7. L221 – Given the reduced hemispheric OH difference improves the simulation
again the question has to be asked why this wasn't used as the standard configuration?

**As noted above, one of our goals is to understand the impacts of model diversity. We**
**therefore thought it best to use an OH distribution similar to what many CCMs, including**
**GEOS when run as a CCM, produce. We now explain this in Section 2.1:**
**"This modification is designed to make our results more applicable to understanding the**
**impacts inter-model differences in Cl, since it makes our OH distribution more consistent**
**by that produced by many CCMs."**

P7. L236-237 – How large would the correction to the geologic source have to be to
improve the inter-hemispheric gradient? It seems that the inter-hemispheric gradient
is relatively unperturbed by the different Cl fields, and more likely a result of source
differences or a change in the interhemispheric OH ratio. The simulation that best
approximates the inter-hemispheric gradient is SimWet. Doesn't this point to accurate
source signatures being of primary importance in determining this gradient, rather than
the Cl field?

**It is beyond the scope of this study to infer the optimum geologic source. Accurate estimates of source magnitudes and signatures are certainly important, and the purpose of this section is indeed to acknowledge that importance. However, we have also shown a significant impact from Cl. Table 3 shows that SimMBL alters the Jan. interhemispheric gradient more than SimWet. Consequently, both sources and sinks are uncertain levers on the interhemispheric gradient.**

P7. L239-240 – Would it not be best to include SimWet description in section 2.2? Also it would be useful to give more details of this simulation. By how much does the wetland source signature change versus the single value simulation?

**We add a brief description, including the global mean signature, to Section 2.2 as suggested:**
**"We conduct an additional sensitivity study, SimWet, to illustrate the role of spatial variation in the isotopic source signature. SimWet parallels SimStd, but the isotopic composition of the wetland source uses spatial variation from Ganesan et al (2018). The global mean source signature of the wetland emissions remains -60‰."**

P8. L265 - "We find that the NH Cl maximum acts to flatten the interhemispheric gradient...". But this isn't the Cl distribution alone that has this effect, as both the OH sink magnitude and fractionation changed. Based on the sensitivity cases shown one cannot conclude that this is down to Cl alone. The same applies for conclusions related to SimMBL.

**We now mention the OH change as well:**
**"We find that the strong NH Cl maximum, along with the resulting reduction in OH fractionation required to maintain consistency with observations, acts to flatten the interhemispheric gradient of $\delta^{13}C$"**

P8. L271 - "...but combining it with the Cantrell et al (1990) value would lead to an overestimate"
Would it? This isn't shown. Changing the fractionating effect of OH between sensitivity cases without showing the evidence for why this is the most appropriate action seems odd.

**We looked at this in the SimTomB simulation, which uses the Cantrell et al (1990) value, and found it quickly becomes too heavy compared to observations. We now mention this is in Section 2.2:**
**"This simulation becomes too heavy compared to observations, justifying the need to change $\alpha_{OH}$ in the main SimTom simulation."**

P8. L272 – What about the uncertainty in the fractionating effect of other sinks such as soil absorption and Cl. Are they of a similar scale to the uncertainty in OH?

**It is true that better constraints on any portion of the methane budget are helpful, but we focus here on the OH fractionation value because it is a very strong hammer on the mean isotopic ratios, and not all of its values are consistent with all the Cl values simulated by global models.**

P8. L272-275 - "The choice of Cl field thus strongly impacts..." Yes, but by how much? Under these different assumptions about Cl how different would the source mixtures end up being?

**We use our new analysis, described earlier, to make this quantitative. We added the following to the conclusions:**
**"Each percent increase in the amount of $CH_4$ loss occurring by reaction with Cl increases global mean surface $\delta^{13}C$ of $CH_4$ by approximately 0.5‰. This relationship can be used to estimate the impact on methane's isotopic values from future model simulations of Cl."**

Technical comments:
P. 3 L 83 – GMI is used before being defined

**Fixed**

Fig. 4 – I appreciate the scales probably need to be different, but why do the panels use different color maps?

**We updated the figure to use the same color maps.**

Fig 8 and 9 – Units omitted on panels b and c in both figures.

**We added the units.**

---

## Author Response (AR2)

**Response to Referee #2**

**The original comments are in regular text and our responses are in bold.**

The authors have made good efforts to address the points raised in the review. The authors now present a quantitative conclusion and provide more justification for various decisions that have been made in setting up their simulations. The additional sensitivity tests that are now provided help to address my earlier concerns. As a result, the manuscript has been improved although it could further benefit from addressing the following minor points:

**We thank the referee for the thoughtful review and respond to specific comments below.**

Specific Comments:

P. 5 L.175-176 "These changes are small compared to the uncertainty in global OH." I think a reference is needed here, perhaps Rigby et al. (2017) or Turner et al. (2017).

**We add a reference to Rigby et al. (2017).**

P.7 L228-229 – Thank you for providing this new Fig 6. which is certainly useful, but could you please provide a brief explanation for why the isotope ratio is overestimated at BRW in contrast to other sites. Is it likely to be due to an incorrect latitudinal distribution of sinks, sources or even transport errors?

**Most of our simulations underestimate the latitudinal gradient of $\delta^{13}CH_4$, leading to the overestimate at BRW compared to the other sites, even though they reproduce well the interhemispheric gradient of total $CH_4$. While multiple factors may contribute, the SimWet simulation, which uses spatially varying isotopic ratios for wetland emissions, gives a reasonable reproduction of the $\delta^{13}CH_4$ observations at both BRW and the southernmost site, SPO. This suggests that latitudinal variability in the isotopic source ratios, which is absent in the other simulations, contributes to the bias at BRW compared to the other stations. We add the following text to Section 3.4:**

**"SimWet is better able to simultaneously match the $\delta^{13}C$-$CH_4$ observations at both the northernmost (BRW) and southernmost (SPO) sites shown in Fig. 6 than the other simulations, even though all simulations reproduce the latitudinal distribution of $CH_4$ well (Fig. 4). This highlights the importance of spatially varying isotopic ratios for the $\delta^{13}C$-$CH_4$ distribution."**

Technical Corrections:

P. 3 L 100 – Missing 'of' in "understanding the impacts [of] inter-model differences"

**Fixed**

P. 5 L180 – "which uses the same Cl field as Cl but retains". Is this meant to say "SimTOM" in place of the second "Cl" perhaps?

**Yes, we changed it to "SimTom".**

[revised manuscript text omitted]